# Uncovering a novel function of the CCR4-NOT complex in phytochrome A-mediated light signalling in plants

Philipp Schwenk[1,2], David J Sheerin[1], Jathish Ponnu[3], Anne-Marie Staudt[1], Klara L Lesch[1,2,4], Elisabeth Lichtenberg[1], Katalin F Medzihradszky[5], Ute Hoecker[3], Eva Klement[5], András Viczián[6], Andreas Hiltbrunner[1,7]*

[1]Institute of Biology II, Faculty of Biology, University of Freiburg, Freiburg, Germany; [2]Spemann Graduate School of Biology and Medicine (SGBM), University of Freiburg, Freiburg, Germany; [3]Institute for Plant Sciences and Cluster of Excellence on Plant Sciences (CEPLAS), University of Cologne, Cologne, Germany; [4]Internal Medicine IV, Department of Medicine, Medical Center, University of Freiburg, Freiburg, Germany; [5]Laboratory of Proteomics Research, Biological Research Centre, Szeged, Hungary; [6]Institute of Plant Biology, Biological Research Centre, Szeged, Hungary; [7]Signalling Research Centres BIOSS and CIBSS, University of Freiburg, Freiburg, Germany

*For correspondence:
andreas.hiltbrunner@biologie.uni-freiburg.de

**Competing interests:** The authors declare that no competing interests exist.

**Abstract** Phytochromes are photoreceptors regulating growth and development in plants. Using the model plant Arabidopsis, we identified a novel signalling pathway downstream of the far-red light-sensing phytochrome, phyA, that depends on the highly conserved CCR4-NOT complex. CCR4-NOT is integral to RNA metabolism in yeast and animals, but its function in plants is largely unknown. NOT9B, an Arabidopsis homologue of human CNOT9, is a component of the CCR4-NOT complex, and acts as negative regulator of phyA-specific light signalling when bound to NOT1, the scaffold protein of the complex. Light-activated phyA interacts with and displaces NOT9B from NOT1, suggesting a potential mechanism for light signalling through CCR4-NOT. ARGONAUTE 1 and proteins involved in splicing associate with NOT9B and we show that NOT9B is required for specific phyA-dependent alternative splicing events. Furthermore, association with nuclear localised ARGONAUTE 1 raises the possibility that NOT9B and CCR4-NOT are involved in phyA-modulated gene expression.

## Introduction

As sessile organisms, plants constantly monitor their surrounding and adapt growth and development to changes in the ambient environment. Light is a key factor for plants that promotes, for instance, seed germination, early seedling development, and transition to flowering (*Paik and Huq, 2019*). To sense the spectral composition and measure the light intensity in the environment, plants have evolved a set of photoreceptors sensitive to wavelengths in the UV-B, blue (B), red (R), and far-red (FR) light range of the spectrum. Phytochromes are receptors for R and FR light. They are synthesised in the inactive Pr state and convert to the physiologically active Pfr state upon absorbing light (*Legris et al., 2019*). Pfr can then revert to the Pr state either by absorption of light or by thermal relaxation. This toggle-switch like behaviour enables phytochromes to sense the red:far-red light ratio in the environment, which is key to distinguish sunlight from canopy shade.

PhyA and phyB are the primary phytochromes in seed plants. PhyB has a dominant function as R light receptor, while responses induced by FR light depend on phyA (*Legris et al., 2019*).

**eLife digest** Place a seedling on a windowsill, and soon you will notice the fragile stem bending towards the glass to soak in the sun and optimize its growth. Plants can 'sense' light thanks to specialized photoreceptor molecules: for instance, the phytochrome A is responsible for detecting weak and 'far-red' light from the very edge of the visible spectrum. Once the phytochrome has been activated, this message is relayed to the rest of the plant through an intricate process that requires other molecules.

The CCR4-NOT protein complex is vital for all plants, animals and fungi, suggesting that it was already present in early life forms. Here, Schwenk et al. examine whether CCR4-NOT could have acquired a new role in plants to help them respond to far-red light.

Scanning the genetic information of the plant model *Arabidopsis thaliana* revealed that the gene encoding the NOT9 subunit of CCR4-NOT had been duplicated in plants during evolution. NOT9B, the protein that the new copy codes for, has a docking site that can attach to both phytochrome A and CCR4-NOT. When NOT9B binds phytochrome A, it is released from the CCR4-NOT complex: this could trigger a cascade of reactions that ultimately changes how *A. thaliana* responds to far-red light. Plants that had not enough or too much NOT9B were respectively more or less responsive to that type of light, showing that the duplication of the gene coding for this subunit had helped plants respond to certain types of light.

The findings by Schwenk et al. illustrate how existing structures can be repurposed during evolution to carry new roles. They also provide a deeper understanding of how plants optimize their growth, a useful piece of information in a world where most people rely on crops as their main source of nutrients.

Phytochromes are located to the cytosol in the inactive state and translocate into the nucleus upon conversion to the active Pfr state (*Klose et al., 2015*). Two comparably well-characterised signalling pathways link light-activation of phytochromes to regulation of gene expression. The PHYTO-CHROME INTERACTING FACTORs (PIFs) are bHLH transcription factors that suppress photomorphogenesis, and in parallel the CONSTITUTIVELY PHOTOMORPHOGENIC 1/SUPPRESSOR OF PHYA-105 (COP1/SPA) E3 ubiquitin ligase complex targets positive factors of light signalling for degradation by the 26S proteasome (*Legris et al., 2019*). Light-activated phytochromes inhibit PIF and COP1/SPA action, and thereby regulate gene expression in response to light. In addition to these well-established light signalling components, also factors involved in splicing or microRNA biogenesis modulate light responses.

Mutants with reduced activity of DICER-LIKE 1 (DCL1), SERRATE (SE), and ARGONAUTE 1 (AGO1), or lacking functional HYPONASTIC LEAVES 1 (HYL1) or HUA ENHANCER 1 (HEN1) are hypersensitive to light (*Achkar et al., 2018*; *Sun et al., 2018*; *Cho et al., 2014*; *Tsai et al., 2014*; *Sorin et al., 2005*) and functional interactions with the classical light signalling components COP1, HY5, and PIF4 have been described (*Sun et al., 2018*; *Cho et al., 2014*; *Tsai et al., 2014*). Mutants with defects in miRNA biogenesis or action often have pleiotropic phenotypes.

Several components of light signalling pathways and the circadian clock are subject to alternative splicing (*Tognacca et al., 2019*; *Hartmann et al., 2016*; *Mancini et al., 2016*; *Shikata et al., 2014*). SPLICING FACTOR FOR PHYTOCHROME SIGNALLING (SFPS) and REDUCED RED-LIGHT RESPONSES IN CRY1 CRY2 BACKGROUND (RRC1) are related to the human splicing factor SR140 and SPF45, respectively (*Xin et al., 2017*; *Shikata et al., 2012*). They interact with each other, directly bind phyB and also associate with pre-mRNAs in vivo (*Xin et al., 2019*; *Xin et al., 2017*). RRC1 regulates thousands of splicing events. Therefore, it is not surprising that *rrc1* loss-of-function mutants are lethal; mutants lacking functional SFPS are viable but show altered splicing of hundreds of transcripts (*Xin et al., 2019*). Both *sfps* and hypomorphic *rrc1* mutants have general defects in light signalling (*Xin et al., 2019*; *Xin et al., 2017*; *Shikata et al., 2012*). Light conditions that do not activate photosynthesis, such as monochromatic FR light or short light pulses, can modulate alternative splicing through photosensory photoreceptors, including phytochromes (*Mancini et al., 2016*). However, many light-regulated splicing events are not affected in photoreceptor mutants and can

be induced by exogenous sugar supply, indicating that they are connected to photosynthetic activity (*Hartmann et al., 2016*; *Mancini et al., 2016*).

Recent findings describe the CARBON CATABOLITE REPRESSION 4-NEGATIVE ON TATA-LESS (CCR4-NOT) complex as a major player in mRNA metabolism in eukaryotes (*Collart, 2016*; *Ukleja et al., 2016*; *Villanyi and Collart, 2015*). The complex consists of several evolutionary conserved subunits that assemble with the scaffold protein NOT1. These subunits have deadenylase (CCR4 ASSOCIATED FACTOR 1 [CAF1] and CCR4) or E3 ubiquitin ligase (NOT4) activity, or are transcriptionally active (NOT2) (*Collart and Panasenko, 2017*; *Ukleja et al., 2016*; *Villanyi and Collart, 2016*). Another highly conserved subunit of the complex is NOT9 (also called REQUIRED FOR CELL DIFFERENTIATION 1 [RQCD1], CAF40, CNOT9), which in animals spans the bridge between the RNA-induced silencing complex (RISC) and CCR4-NOT (*Chen et al., 2014*; *Mathys et al., 2014*). The CCR4-NOT complex controls gene expression at all steps from transcription in the nucleus to translation and mRNA degradation in the cytosol (*Collart, 2016*).

Although extensively studied in yeast and animals, the CCR4-NOT complex was only recently shown to exist in plants (*Zhou et al., 2020*; *Arae et al., 2019*). The full complex regulates RNA-directed DNA methylation in Arabidopsis (*Zhou et al., 2020*), and the two NOT2 homologues, NOT2A and NOT2B, promote polymerase II-dependent transcription and contribute to miRNA biogenesis (*Wang et al., 2013*). Similar to the CCR4-CAF1 module in yeast and animals, Arabidopsis homologues of CCR4-CAF1 have mRNA deadenylase activity and play a role as integrators of environmental stresses (*Suzuki et al., 2015*; *Walley et al., 2010*; *Liang et al., 2009*). However, the function of other components of the complex and their relevance for specific signalling pathways is largely unknown. Mutants lacking the scaffold protein NOT1 are lethal in plants, animals, and yeast (*Zhou et al., 2020*; *Motomura et al., 2020*; *Pereira et al., 2020*; *Ito et al., 2011*; *DeBella et al., 2006*; *Maillet et al., 2000*).

There are two close homologues of CNOT9/CAF40/RQCD1 in Arabidopsis, NOT9A and NOT9B, and a more distantly related protein, NOT9C (*Figure 1—figure supplement 1*). In this study, we show that NOT9B binds to light activated phyA and acts as repressor of early seedling development during the dark-to-light transition. We describe an unexpected role of the Arabidopsis CCR4-NOT complex in light signalling and show that NOT9B – despite being a component of this evolutionary conserved complex – has a very specific role in phyA-regulated photomorphogenesis.

## Results

### NOT9B is a novel phytochrome A interacting protein

In a yeast-two-hybrid (Y2H) screen for novel phyA-interacting proteins, we identified NOT9B, an Arabidopsis homologue of CNOT9/CAF40 from yeast and animals. As a primary validation, we performed yeast-two-hybrid (Y2H) growth assays (*Figure 1A*). Irrespective of whether phyA was in the active Pfr or inactive Pr state, we found interaction with NOT9B in yeast, while FAR-RED ELONGATED HYPOCOTYL 1 (FHY1), a control protein known to specifically interact with light-activated phyA in the Y2H system (*Hiltbrunner et al., 2005*), only bound phyA Pfr but not Pr. Further Y2H experiments revealed that the N-terminal 406 amino acids of phyA are sufficient for binding NOT9B (*Figure 1—figure supplement 2*). NOT9A, the closest homologue of NOT9B in Arabidopsis, and the more distantly related NOT9C did not interact with phyA in the Y2H system under high stringency conditions and were therefore not further tested (*Figure 1—figure supplements 1*, *3A and B*). Interaction of NOT9B with phyA was also observed in co-immunoprecipitation (CoIP) assays from HEK293T cells transfected with plasmids coding for FLAG-myc-mCherry-NOT9B and phyA-GFP (*Figure 1—figure supplement 4*).

In order to verify the interaction in planta, we performed CoIP assays using extracts from transiently transformed tobacco plants. We could co-precipitate phyA-NLS-3×HA with HA-YFP-NOT9B using anti-GFP traps (*Figure 1B*). In addition, YFP fluorescence lifetime was significantly reduced in FRET/FLIM experiments with transiently transformed leek cells co-expressing HA-YFP-NOT9B and phyA-NLS-tagRFP compared to cells expressing HA-YFP-NOT9B and tagRFP only (*Figure 1C*). Finally, we used stable transgenic Arabidopsis lines expressing *p35S:HA-YFP-NOT9B* in Col-0 background for CoIP assays to verify the phyA/NOT9B interaction and test for Pfr dependence. Seedlings were exposed to FR light for 5 hr to facilitate nuclear import of phyA and subsequently treated for 5

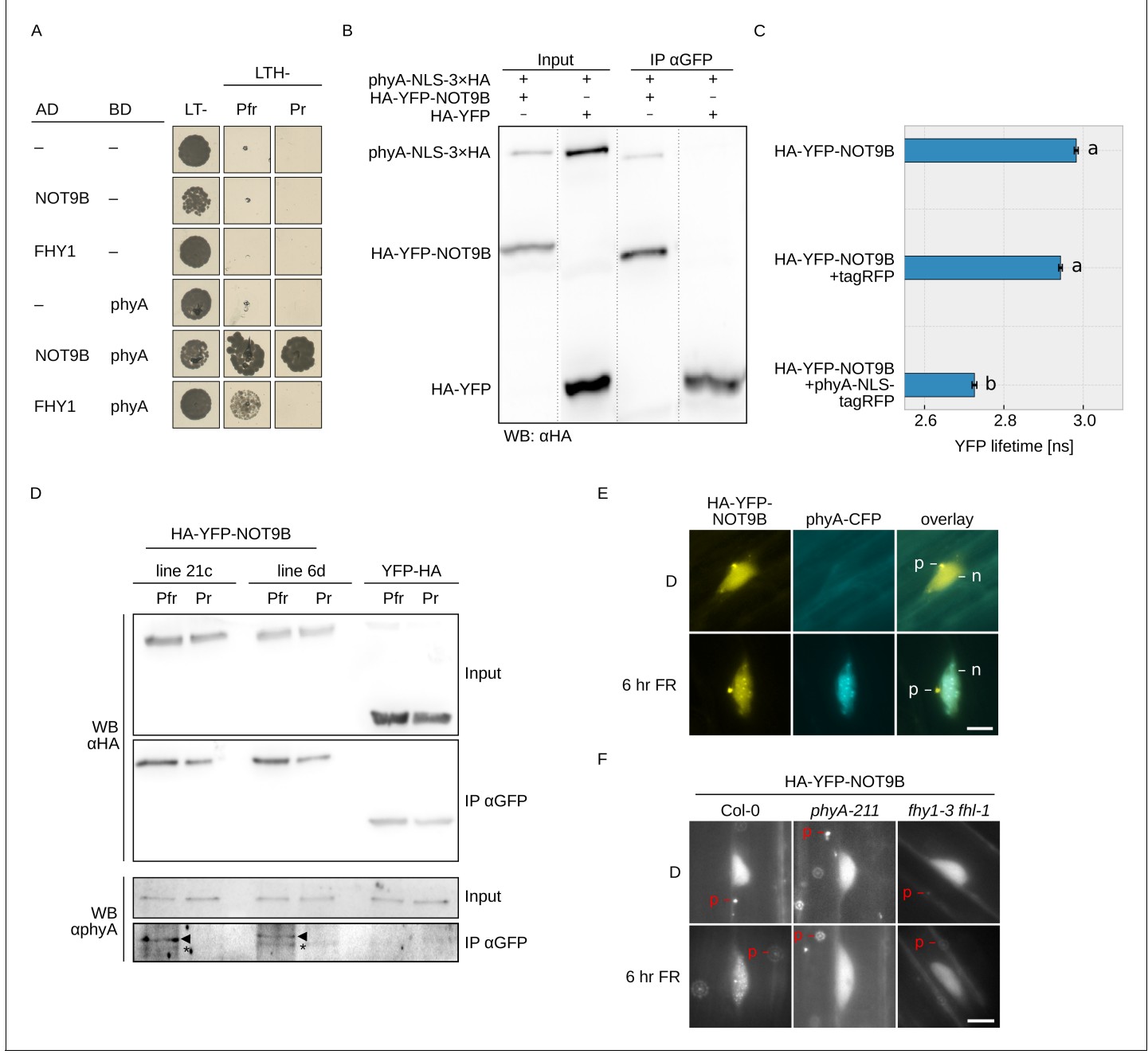

**Figure 1.** NOT9B interacts with phytochrome A. (**A**) Y2H growth assay. AH109 cells were transformed with plasmids coding for indicated AD and BD fusion proteins and grown on CSM LTH- plates supplemented with 20 μM phycocyanobilin and 5 mM 3-AT to test for interaction. Growth on CSM LT- plates was used as transformation control. Plates were incubated in R or FR light to convert phyA to Pfr and Pr, respectively. (**B**) CoIP from transiently transformed tobacco leaves. Four-week-old tobacco leaves were infiltrated with Agrobacteria transformed with plasmids coding for *p35S:PHYA-NLS-3×HA* and either *p35S:HA-YFP-NOT9B* or *p35S:HA-YFP*. Plants were kept in the dark and exposed to W light for 5 min. Total soluble protein was then used for CoIP with αGFP magnetic beads. Input and eluate fractions were analysed by SDS-PAGE and immunoblotting with αHA antibodies. (**C**) FRET-FLIM experiment in leek cells. Leek cells were bombarded with plasmids coding for *p35S:HA-YFP-NOT9B*, *p35S:PHYA-NLS-tagRFP*, or *p35S:tagRFP*. FRET-FLIM was quantified for the YFP-tagRFP FRET pair. Mean YFP fluorescence life time of >15 replicates (±SE) is shown. Letters indicate levels of significance as determined by one-way ANOVA followed by post-hoc Tukeys HSD test; p<0.05. (**D**) CoIP from stable transgenic Arabidopsis lines. Four-day-old dark-grown Arabidopsis seedlings expressing either *p35S:HA-YFP-NOT9B* or *p35S:YFP-HA* were exposed to FR light for 5 hr followed by 5 min R (Pfr) or long-wavelength FR light (Pr). Total soluble protein was then used for CoIP with αGFP magnetic beads. Input and eluate fractions were analysed by SDS-PAGE and immunoblotting with αHA and αphyA antibodies. *, unspecific band; ▶, phyA. (**E** and **F**) Subcellular localisation of NOT9B. Four-day-old dark-grown Arabidopsis seedlings co-expressing *p35S:HA-YFP-NOT9B* and *pPHYA:PHYA-CFP* (**E**) or expressing *p35S:HA-YFP-NOT9B* in

*Figure 1 continued on next page*

*Figure 1 continued*

Col-0, *phyA-211*, or *fhy1-3 fhl-1* background (F) were irradiated with FR light for 6 hr or kept in the dark (D) and analysed by epifluorescence microscopy. Scale bar represents 5 μm. p, p-body; n, nucleus.

The online version of this article includes the following figure supplement(s) for figure 1:

**Figure supplement 1.** Sequence alignment for human CNOT9 and Arabidopsis NOT9 proteins.
**Figure supplement 2.** NOT9B binds to the N-terminal half of phyA.
**Figure supplement 3.** NOT9B homologues do not affect phyA signalling.
**Figure supplement 4.** NOT9B co-precipitates with phyA in HEK293T cells.
**Figure supplement 5.** NOT9B forms photobodies in FR but not R light.

min with either R light (660 nm) or long-wavelength FR light (760 nm) to convert phyA predominantly to the Pfr or Pr state, respectively. The amount of phyA that co-purified with NOT9B was substantially higher when it was in the active Pfr state compared to the inactive Pr state (*Figure 1D*). For unknown reasons, the interaction in the Y2H system was not affected by the Pr/Pfr state of phyA, however we have previously observed that the Pfr-dependency of phyA interactions can be less pronounced in the Y2H system than in CoIP assays (*Enderle et al., 2017*; *Sheerin et al., 2015*).

Light-activated phyA translocates from the cytosol into the nucleus and forms subnuclear structures termed photobodies (*Klose et al., 2015*). Consistent with the interaction of phyA and NOT9B, HA-YFP-NOT9B colocalised with phyA-CFP in photobodies following exposure to FR light (*Figure 1E*), but did not form photobodies in R light (*Figure 1—figure supplement 5*). Furthermore, recruitment of HA-YFP-NOT9B into photobodies was abolished in *phyA-211*, a mutant lacking functional phyA (*Reed et al., 1994*), and *fhy1-3 fhy1-like-1* (*fhl-1*), which is impaired in phyA nuclear transport (*Figure 1F*; *Hiltbrunner et al., 2006*). In contrast to NOT9B, NOT9A and NOT9C did not form detectable photobodies, consistent with the observations from interaction assays that only NOT9B binds to phyA (*Figure 1—figure supplement 3C*). Overall, we conclude that NOT9B interacts with phyA and that interaction in planta is enhanced when phyA is in the active Pfr state.

## NOT9B is a negative regulator of phyA-mediated light signalling

To investigate the potential role of NOT9B in light signalling, we isolated two independent *not9b* T-DNA insertion alleles. *not9b-2* carries a T-DNA insertion in the third exon of *NOT9B* and lacks detectable expression, whereas the *not9b-1* allele has an insertion in the 3' UTR, which results in reduced transcript levels (*Figure 2A,B*). Both *not9b* mutant alleles are hypersensitive to FR light with regard to inhibition of hypocotyl growth, while hypocotyl length of *not9b* mutants is indistinguishable from the wildtype when grown in the dark or either R or B light (*Figure 2C,D*, *Figure 2—figure supplement 1A,B*). The *not9b-2 phyB-9* double mutant is as hypersensitive to FR light as the *not9b-2* single mutant but lacks a response to R light (*Figure 2—figure supplement 1C,D*). In contrast, *not9b-2 phyA-211* is fully insensitive to FR light, similar to *phyA-211*, but behaves like the wildtype when grown in R light (*Figure 2—figure supplement 1C,D*). Expression of LUC-NOT9B under the control of the native promoter complements the *not9b-2* mutant phenotype, confirming that lack of functional NOT9B in *not9b-2* is responsible for the increased sensitivity to FR light (*Figure 2E*).

To further investigate a function of NOT9B in phyA-dependent light signalling, we examined HA-YFP-NOT9B overexpression lines (*p35S:HA-YFP-NOT9B* in Col-0, NOT9Box). NOT9Box lines are hyposensitive to FR light in a dose-dependent manner (*Figure 2D,F*, *Figure 2—figure supplement 2*), whereas overexpression of NOT9A and NOT9C did not affect hypocotyl growth in FR light (*Figure 2—figure supplement 3A,B*). Similar to *not9b*, NOT9Box lines are phenotypically indistinguishable from the wildtype when grown in the dark or R light (*Figure 2D*, *Figure 2—figure supplement 1E*). NOT9Box does not further increase hypocotyl growth of *phyA-211* in FR light, and the effect of NOT9Box is not affected by the absence of functional phyB (*Figure 2—figure supplement 1F*). Furthermore, we found that NOT9Box seedlings expressing constitutively nuclear localised phyA are equally hyposensitive to FR light as the NOT9Box parental line, in which phyA nuclear transport depends on FHY1 and FHL (*Hiltbrunner et al., 2006*), and that phyA-CFP still accumulates in the nucleus in NOT9Box background (*Figure 1E*, *Figure 2—figure supplement 1G*). Therefore,

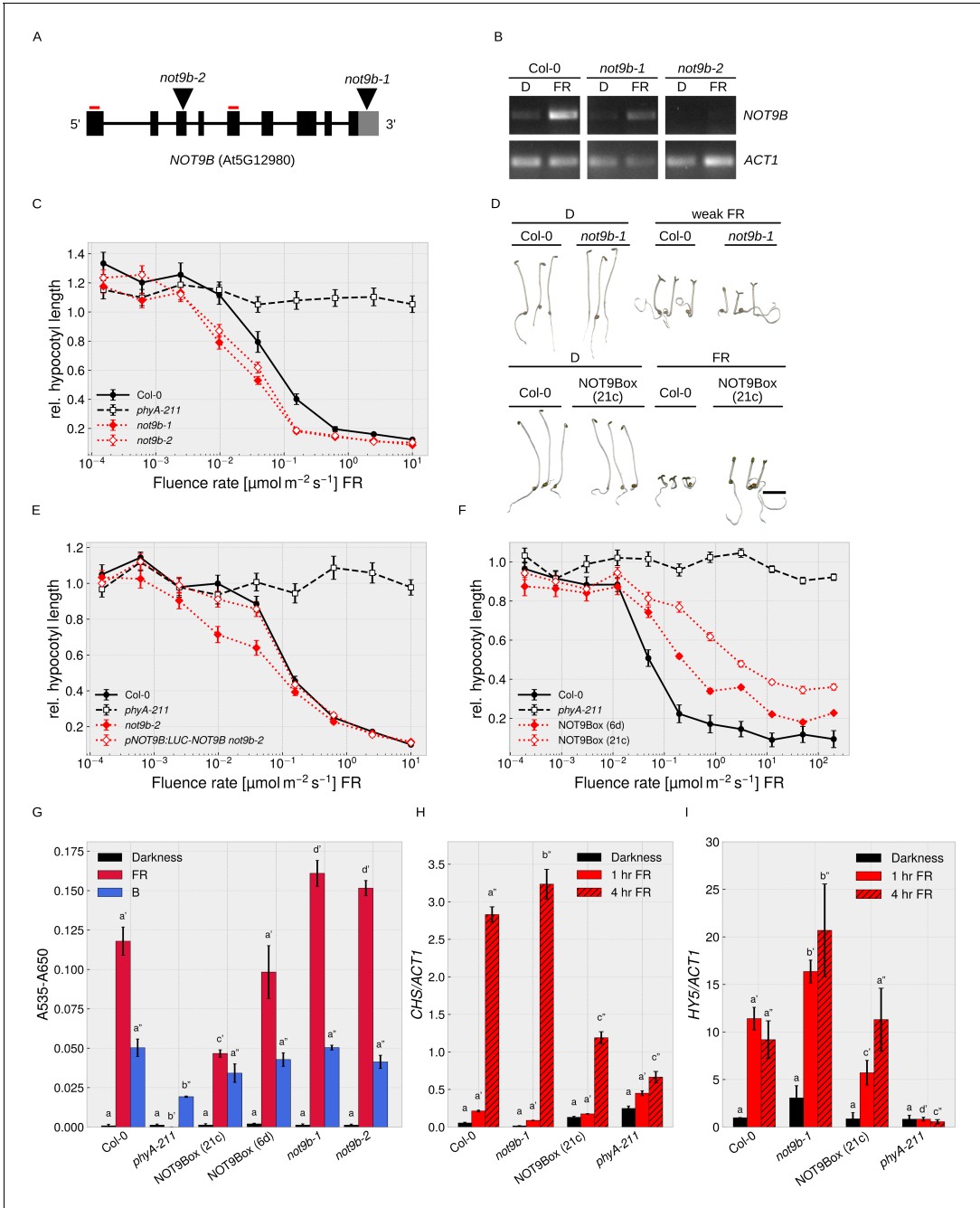

**Figure 2.** NOT9B is a negative regulator of phyA-mediated light signalling. (**A**) *NOT9B* T-DNA insertion mutants. Schematic drawing of the *NOT9B* genomic locus. T-DNA insertions in *not9b-1* and *not9b-2* are shown as triangles and primers used for RT-PCR in (**B**) as red lines. Black boxes, exons; black lines, introns; grey box, 3' UTR. (**B**) RT-PCR for *not9b* alleles. Four-day-old dark-grown Col-0, *not9b-1*, and *not9b-2* seedlings were exposed to FR light for 24 hr (FR) or kept in the dark. Total RNA was extracted and *NOT9B* and *ACT1* transcripts were detected by RT-PCR and agarose gel electrophoresis. Positions of primers used to detect *NOT9B* are indicated in (**A**). *ACT1* was used as reference gene. (**C**), (**E**), (**F**) Fluence rate response curves for inhibition of hypocotyl growth. Seedlings of the indicated genotypes were grown for 4 days at different fluence rates of FR light or in the dark. Mean hypocotyl length (±SE) of 20 seedlings relative to dark-grown seedlings is shown. NOT9Box 6d and 21c are two independent lines expressing *p35S:HA-YFP-NOT9B*. Expression of *pNOT9B:LUC-NOT9B* was used for complementation of *not9-2*. (**D**) Phenotype of *not9b-1* and NOT9Box seedlings. Seedlings were grown for 4 days in the dark or in FR light (FR, 10 µmol m$^{-2}$ s$^{-1}$; weak FR, 0.05 µmol m$^{-2}$ s$^{-1}$). Scale bar represents 2 mm. (**G**) Quantification of anthocyanin levels in *not9b-1* and NOT9Box seedlings. Seedlings of indicated genotypes were grown for 4 days in the dark (D), FR, or B light on ½ MS supplemented with 1.5% sucrose. Anthocyanin was extracted from 25 seedlings per genotype/condition and A$_{535}$-A$_{650}$ was measured. Bars represent biological triplicates (± SD). (**G**)-(**I**) Letters indicate levels of significance as determined by one-way ANOVA followed by post-hoc Tukeys HSD test; p<0.05. (**H** and **I**) Quantification of *CHS* and *HY5* transcript levels in *not9b-1* and NOT9Box seedlings. Four-day-old dark-

*Figure 2 continued*

grown seedlings were exposed to FR light for 1 or 4 hr or kept in the dark. Total RNA was extracted and *CHS*, *HY5,* and *ACT1* transcript levels were quantified by qRT-PCR. Bars represent biological triplicates (± SD).

The online version of this article includes the following figure supplement(s) for figure 2:

**Figure supplement 1.** Fluence rate response curves for inhibition of hypocotyl growth and seed germination experiment.
**Figure supplement 2.** Western blot analysis of protein levels.
**Figure supplement 3.** NOT9B homologues do not affect phyA signalling.
**Figure supplement 4.** Expression pattern and light-regulation of NOT9B transcript and protein levels.

overexpression of NOT9B does not interfere with phyA nuclear transport but possibly affects phyA downstream signalling events in the nucleus.

In addition to hypocotyl growth, we investigated anthocyanin biosynthesis and seed germination in *not9b* mutants and NOT9Box lines. Compared to the wildtype, NOT9Box lines accumulate reduced amounts of anthocyanin specifically in response to FR but not B light, whereas anthocyanin levels are increased in *not9b* mutant alleles (*Figure 2G*). HY5 and CHS are involved in anthocyanin biosynthesis (*Shin et al., 2007*; *Tepperman et al., 2006*; *Zhou et al., 2005*), and we found that *HY5* and *CHS* expression is downregulated in NOT9Box lines and upregulated in the *not9b-1* mutant (*Figure 2H,I*). Under conditions where phyA promotes seed germination, *not9b* mutant seeds had a slightly higher germination rate than wildtype seeds (*Figure 2—figure supplement 1H*).

In *pNOT9B:GUS* reporter lines, the activity of the *NOT9B* promoter was most prominent in the cotyledons and the upper part of the hypocotyl, which is consistent with a function of *NOT9B* in modulation of phyA-controlled hypocotyl growth (*Figure 2—figure supplement 4A*). *NOT9B* transcript levels were not regulated by light at early time points but increased after 24 hr in W, FR, or B light; in FR light, regulation was dependent on phyA (*Figure 2B*, *Figure 2—figure supplement 4A, B,C*). In seedlings expressing *pNOT9B:LUC-NOT9B*, LUC-NOT9B levels slightly increased after 24 hr exposure to W, R, FR, or B light, while there was no difference compared to dark-grown seedlings at early time points (*Figure 2—figure supplement 4D*).

Overall, we conclude that NOT9B is a negative regulator of early seedling development specifically acting downstream of phyA.

## NOT9B is an evolutionary conserved core component of the CCR4-NOT complex

NOT9B has high sequence similarity to yeast CAF40 and human CNOT9, which are core components of the CCR4-NOT complex (*Figure 3A*, *Figure 1—figure supplement 1*). AP-MS experiments and Y2H assays with truncated NOT1 suggested that a similar complex also exists in plants (*Zhou et al., 2020*; *Arae et al., 2019*). We confirmed interaction of full-length NOT1 with NOT9B, NOT2B, CAF1A, and CAF1B in Y2H assays (*Figure 3B,C*). CNOT9 in humans binds to the DUF3819 domain (DUF) of NOT1 (*Chen et al., 2014*). In Y2H experiments, Arabidopsis NOT9B did interact with NOT1 M, a NOT1 fragment consisting of the DUF and the MIF4G domains, but it did not interact with NOT1 MIF4G fused to GFP, which is similar in size to NOT1 M but lacks the DUF domain (*Figure 3D,E*). NOT1 is 269 kDa in size and tagged full-length NOT1 is difficult to express in plants. In contrast, fragments corresponding to the NOT1 MIF4G and NOT1 M domain express well. Binding of NOT9B to NOT1 M and the DUF domain of NOT1 (NOT1 DUF) was confirmed by CoIP from transiently transformed tobacco leaves. Expression of NOT1 DUF was very low and the protein was not detectable in the input fraction, but we could co-precipitate detectable amounts of NOT1 DUF with NOT9B (*Figure 3F*). Similar to NOT9B, NOT9A did interact with NOT1 in the Y2H system, whereas we did not observe interaction between NOT9C and NOT1 (*Figure 1—figure supplement 3B*).

## Nuclear localised NOT9B is sufficient for modulating phyA signalling

NOT9B has a dual localisation, both nuclear and also forming cytoplasmic structures that are reminiscent of processing bodies (p-bodies) (*Figure 1E,F*, *Figure 1—figure supplement 5*; *Collart, 2016*; *Maldonado-Bonilla, 2014*; *Nissan and Parker, 2008*). Since CCR4-NOT components in yeast and animals are associated with p-bodies, we tested for co-localisation of NOT9B and the established

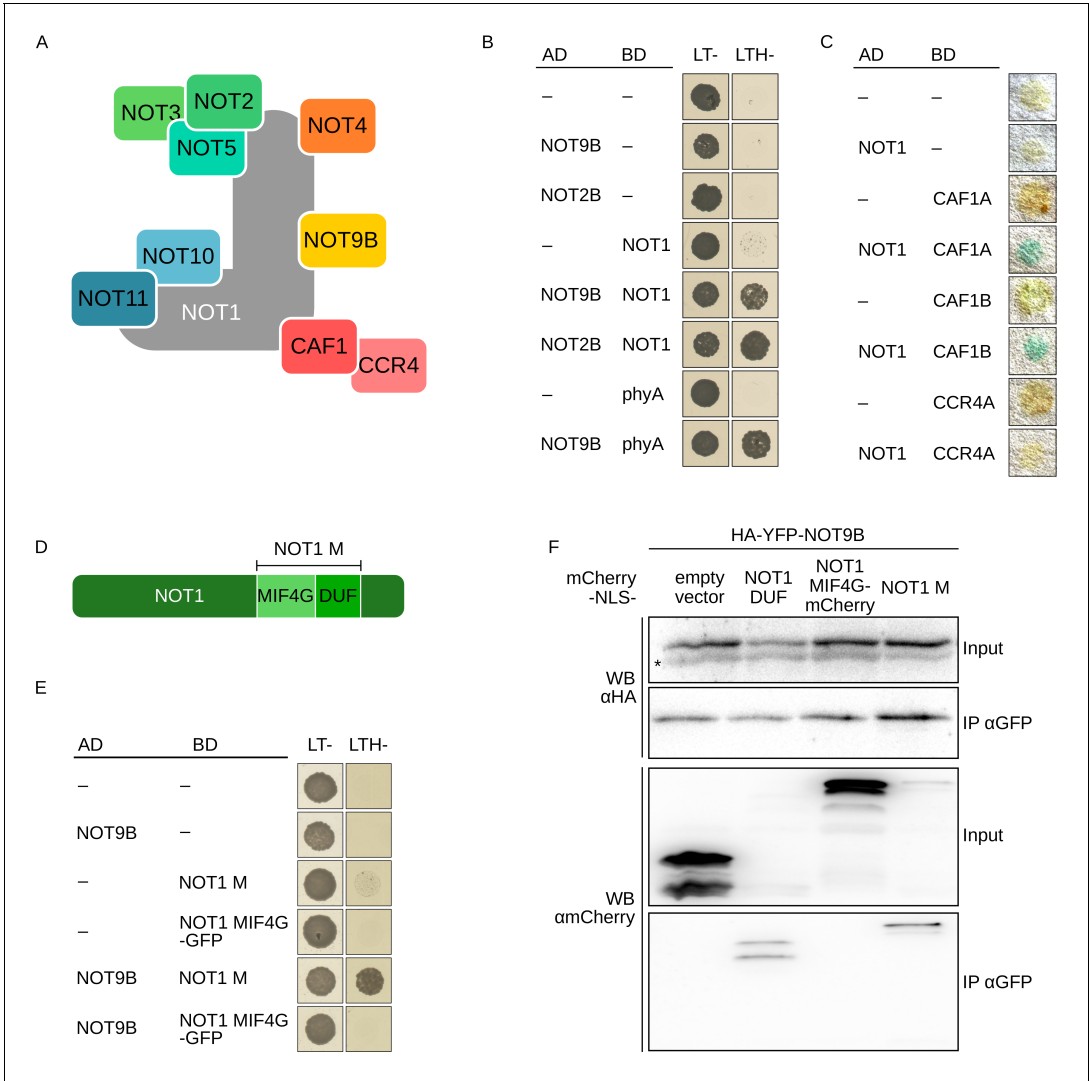

**Figure 3.** NOT9B is part of the CCR4-NOT complex. (**A**) Schematic representation of the yeast/human CCR4-NOT core complex (*Collart, 2016*; *Collart and Panasenko, 2017*). (**B**) Y2H growth assay with components of the CCR4-NOT complex. AH109 cells were transformed with plasmids coding for the indicated AD and BD fusion proteins and grown on CSM LTH- plates supplemented with 25 mM 3-AT to test for interaction. Growth on CSM LT- plates was used as transformation control. (**C**) Y2H filter lift assay. Y190 yeast cells were transformed with plasmids coding the indicated AD and BD fusion proteins and grown on CSM LT- plates. Colonies were lifted onto filter paper and assayed for β-Galactosidase activity, which results in blue staining. (**D**) Schematic representation of NOT1 (*Collart, 2016*; *Collart and Panasenko, 2017*). (**E**) Y2H growth assay with NOT1 fragments. AH109 cells were transformed with plasmids coding for the indicated AD and BD fusion proteins and grown on CSM LTH- plates supplemented with 25 mM 3-AT to test for interaction. (**F**) CoIP from transiently transformed tobacco leaves. Leaves of 4-week-old tobacco plants were infiltrated with Agrobacteria transformed with plasmids coding for HA-YFP-NOT9B and different NOT1 fragments tagged with FLAG-myc-mCherry-NLS (mCherry-NLS). Plants were kept in the dark for 3 days followed by 5 min irradiation with white light. Total soluble protein was extracted, and CoIP was performed using αGFP magnetic beads. Input and eluate fractions were analysed by SDS-PAGE and immunoblotting with αHA and αmCherry antibodies. *, unspecific band.

p-body markers DCP1, XRN4, and AGO1 in plants (*Maldonado-Bonilla, 2014*; *Pomeranz et al., 2010*; *Weber et al., 2008*; *Xu et al., 2006*). CFP-tagged DCP1, XRN4, and AGO1 co-localised with HA-YFP-NOT9B in cytosolic foci in transiently transformed tobacco leaves, and also in stable transgenic Arabidopsis lines we observed colocalisation of DCP1-CFP and HA-YFP-NOT9B in p-bodies (*Figure 4A*, *Figure 4—figure supplement 1A,B,C*).

GW-repeat proteins bind to CNOT9, the human homologue of NOT9B, and possibly contribute to recruitment of the CCR4-NOT complex into p-bodies (*Hicks et al., 2017*; *Mathys et al., 2014*; *Behm-Ansmant et al., 2006*; *Yang et al., 2004*). Structural analyses have shown that TNRC6, a GW-repeat containing protein, binds to CNOT9 through a Trp-binding pocket (*Chen et al., 2014*;

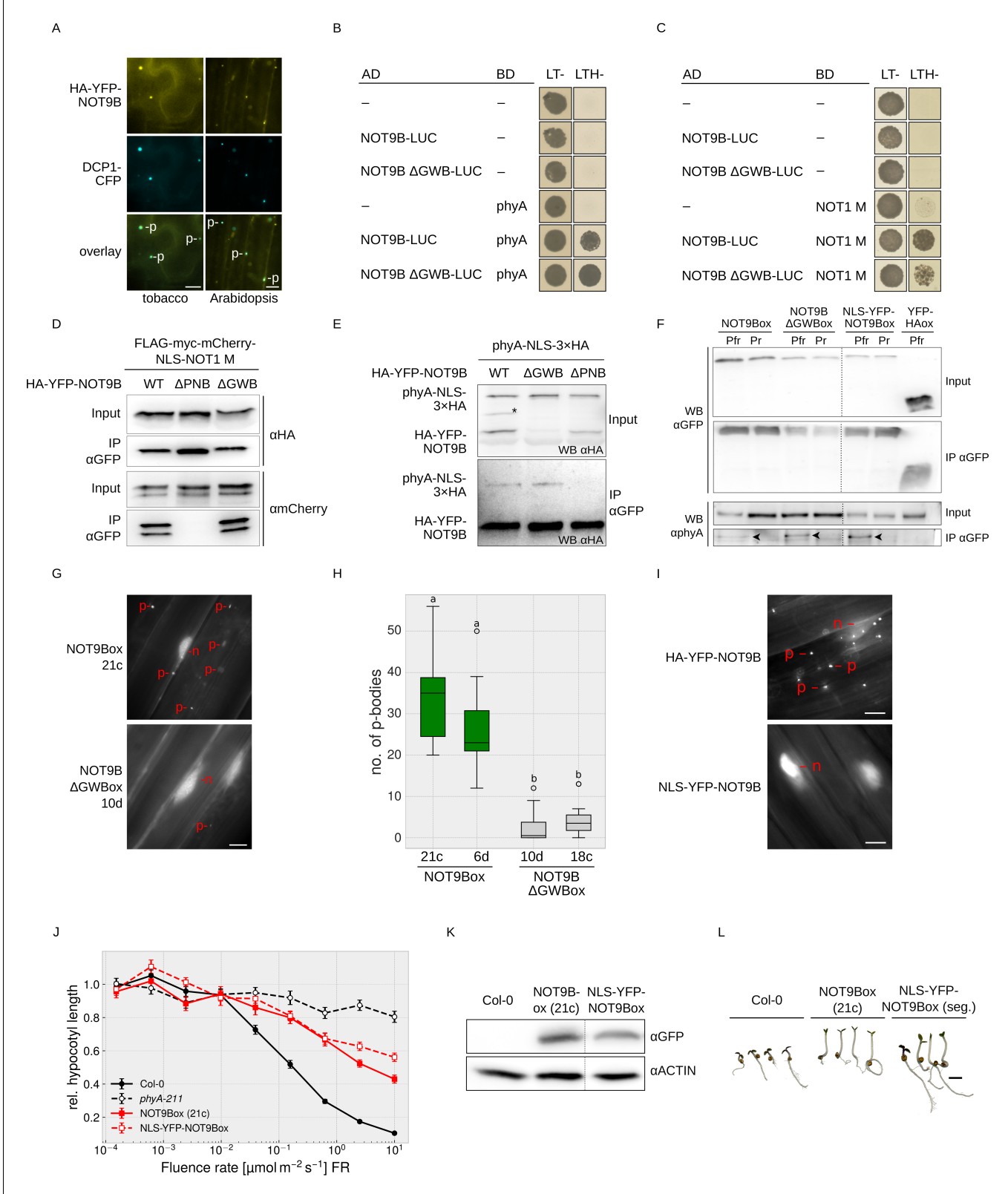

**Figure 4.** NOT9B is a dual-localised protein with both a nuclear and a p-body localised fraction. (**A**) Subcellular localisation of NOT9B. Leaves of 4-week-old tobacco plants were infiltrated with Agrobacteria transformed with plasmids coding for *p35S:HA-YFP-NOT9B* and *p35S:DCP1-CFP*. Plants were then incubated in white light for 3 days. Arabidopsis seedlings co-expressing *p35S:HA-YFP-NOT9B* and *p35S:DCP1-CFP* were grown in the dark for 4 days. Localisation of NOT9B and DCP1 was investigated by epifluorescence microscopy. Scale bars represent 5 μm. p, p-body. An intensity plot
*Figure 4 continued on next page*

Figure 4 continued

for DCP1-CFP and YFP-NOT9B co-expressed in Arabidopsis is shown in *Figure 4—figure supplement 1*. (**B** and **C**) Y2H growth assays. AH109 yeast cells were transformed with plasmids coding for the indicated AD and BD fusion proteins and grown in R light on CSM LTH- plates supplemented with 10 mM 3-AT and 20 µM phycocyanobilin (**B**) or grown on CSM LTH- plates containing 25 mM 3-AT (**C**). CSM LT- plates were used as transformation control. (**D** and **E**) CoIP from transiently transformed tobacco leaves. Leaves of 4-week-old tobacco plants were co-infiltrated with Agrobacteria transformed with plasmids coding for different HA-YFP-NOT9B versions (WT, ΔPNB, ΔGWB) and either *p35S:FLAG-myc-mCherry-NLS-NOT1 M* (**D**) or *p35S:PHYA-NLS-3×HA* (**E**). Plants were kept in the dark for 3 days followed by 5 min irradiation with white light. Total soluble protein was extracted and CoIP was performed using αGFP magnetic beads. Input and eluate fractions were analysed by SDS-PAGE and immunoblotting with αHA (**E**) or αHA and αmCherry antibodies (**D**). *, unspecific band. (**F**) CoIP from transgenic Arabidopsis seedlings. Arabidopsis seedlings expressing *p35S:HA-YFP-NOT9B* (NOT9Box), *p35S:HA-YFP-NOT9B ΔGWB*, *p35S:NLS-YFP-NOT9B*, or *p35S:YFP-HA* (negative control) were grown for four days in the dark and subjected to FR light for 5 hr followed by 5 min of R light (Pfr) or long-wavelength FR light (Pr). Total soluble protein was extracted, and CoIP was performed using αGFP magnetic beads. Input and eluate fractions were analysed by SDS-PAGE and immunoblotting with αGFP and αphyA. Arrowheads point to phyA in the eluate fraction. (**G**) Subcellular localisation of NOT9B ΔGWB. Arabidopsis seedlings expressing *p35S:HA-YFP-NOT9B* (NOT9Box 21c) or *p35S:HA-YFP-NOT9B ΔGWB* (NOT9B ΔGWB 10d) were grown for 4 days in the dark followed by 6 hr in FR light and subjected to epifluorescence microscopy. Scale bar represents 5 µm. p, p-body; n, nucleus. (**H**) Quantification of p-bodies. Seedlings of two independent lines expressing *p35S:HA-YFP-NOT9B* (NOT9Box) or *p35S:HA-YFP-NOT9B ΔGWB* (NOT9B ΔGWBox) were grown as in (**G**). Pictures were taken and number of p-bodies was counted in a 200 × 350 pixel area. Boxplots represent 12 evaluated areas per line. Letters indicate levels of significance as determined by one-way ANOVA followed by post-hoc Tukeys HSD test; p<0.05. (**I**) Subcellular localisation of NLS-YFP-NOT9Box. Four-day-old dark-grown seedlings expressing either *p35S:HA-YFP-NOT9B* or *p35S:NLS-YFP-NOT9B* were analysed by epifluorescence microscopy. Scale bar represents 5 µm. (**J**) Fluence rate response curve for inhibition of hypocotyl growth. Arabidopsis seedlings expressing *p35S:HA-YFP-NOT9B* (NOT9Box) or *p35S:NLS-YFP-NOT9B* (NLS-YFP-NOT9Box) were grown for 4 days in FR light. Mean hypocotyl length (±SE) of 20 seedlings relative to dark-grown seedlings is shown. (**K**) Western blot analysis of protein levels. Total protein was extracted from 4-day-old dark-grown Arabidopsis seedlings expressing *p35S:HA-YFP-NOT9B* (NOT9Box) or *p35S:NLS-YFP-NOT9B* (NLS-YFP-NOT9Box) and analysed by SDS-PAGE and immunoblotting with αGFP and αACTIN antibodies. ACTIN was detected as loading control. (**L**) Phenotype of NLS-YFP-NOT9Box seedlings. NOT9Box seedlings and a segregating population of the *p35S: NLS-YFP-NOT9B* expressing line shown in (**J** and **K**) were grown for 4 days in FR light (FR, 40 µmol $m^{-2}$ $s^{-1}$). Scale bar represents 2 mm.

The online version of this article includes the following figure supplement(s) for figure 4:

**Figure supplement 1.** Colocalisation of NOT9B with p-body markers.
**Figure supplement 2.** Analysis of p-body formation for NOT9B mutants.

---

*Mathys et al., 2014*), which is conserved in NOT9B (*Figure 1—figure supplement 1*). The NOT9B ΔGWB mutant (deficient in <u>GW-b</u>inding; NOT9B R217D A220D R256E A260L) contains amino acid substitutions at positions that are critical for the interaction of human CNOT9 and TNRC6. This mutant is not affected in binding NOT1 and phyA in Y2H assays, which we confirmed by CoIPs from transiently transformed tobacco leaves and stable transgenic Arabidopsis lines (*Figure 4B–F*). However, p-body localisation of HA-YFP-NOT9B ΔGWB in Arabidopsis is strongly reduced compared to NOT9Box, suggesting that proteins associating with NOT9B through the Trp-binding pocket promote assembly of NOT9B into p-bodies (*Figure 4G,H*, *Figure 4—figure supplement 2*).

The dual localisation of NOT9B raises the question whether the nuclear or the cytoplasmic fraction is causing the effect on phyA signalling. Therefore, we generated an Arabidopsis line expressing exclusively nuclear localised NOT9B by adding the strong SV40 NLS (*Figure 4I*). Using CoIP assays, we found that phyA co-precipitates with NLS-YFP-NOT9B in a Pfr dependent manner, indicating that the nuclear localised fraction of NOT9B interacts with light-activated phyA (*Figure 4F*). Seedlings overexpressing constitutively nuclear localised NOT9B are equally hyposensitive to FR light with regard to inhibition of hypocotyl growth as seedlings overexpressing YFP-tagged wildtype NOT9B (*Figure 4J–L*). Therefore, we conclude that the nuclear localised fraction of NOT9B is sufficient for modulation of phyA signalling.

## NOT1 and phyA share a common binding site in NOT9B

The interaction interface of human CNOT1 and CNOT9 has been characterised using co-crystallisation experiments (*Chen et al., 2014*; *Mathys et al., 2014*). Introducing four mutations in CNOT9 disrupted interaction with CNOT1 (*Chen et al., 2014*). The CNOT1 interaction interface is conserved in Arabidopsis NOT9B (*Figure 1—figure supplement 1*), and Y2H growth assays show that NOT9B containing H70A V72A A77Y V85Y substitutions is unable to bind NOT1 M (*Figure 5A*), which we validated by CoIP from transiently transformed tobacco leaves (*Figures 4D* and *5B*). Most interestingly, this NOT9B mutant version is also impaired in phyA binding, as shown by Y2H and CoIP (*Figures 4E* and *5C–E*), therefore we refer to it as NOT9B ΔPNB (defective in <u>p</u>hytochrome and

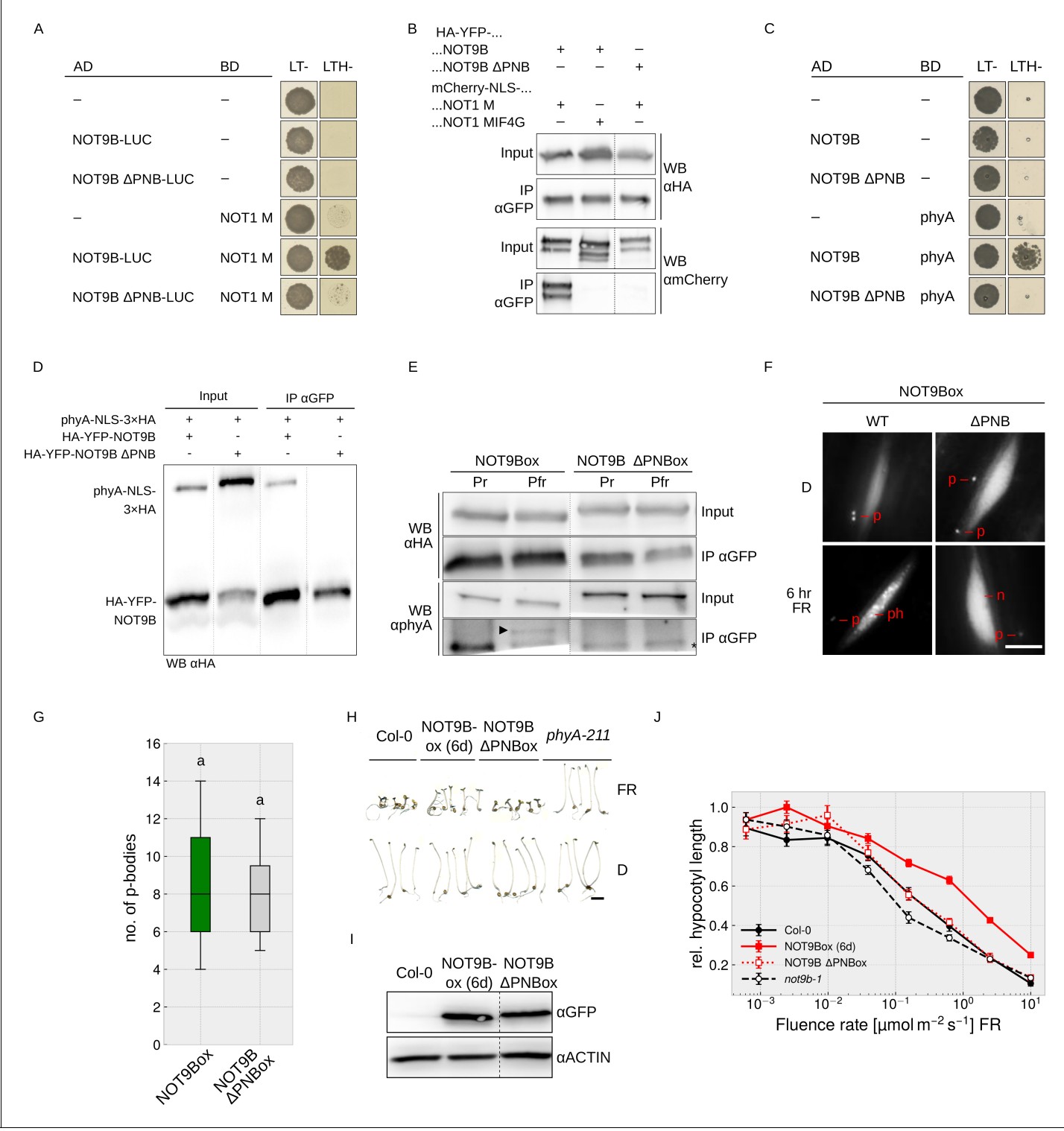

**Figure 5.** Both phyA and NOT1 bind to the PNB site of NOT9B. (**A** and **C**) Y2H growth assays. AH109 yeast cells were transformed with plasmids coding for the indicated AD and BD fusion proteins and grown on CSM LTH- plates containing 25 mM 3-AT (**A**) or grown in R light on CSM LTH- plates supplemented with 10 mM 3-AT and 20 µM phycocyanobilin (**C**). CSM LT- plates were used as transformation control. (**B** and **D**) CoIP from transiently transformed tobacco. Leaves of 4 week-old tobacco plants were co-infiltrated with Agrobacteria transformed with plasmids coding for *p35S:HA-YFP-NOT9B* or *-NOT9B ΔPNB* and *p35S:FLAG-myc-mCherry-NLS-NOT1 M* or *-NOT1 MIF4G-mCherry* (**B**), and *p35S:PHYA-NLS-3×HA* and either *p35S:HA-YFP-NOT9B* or *-NOT9B ΔPNB* (**D**). Plants were kept in the dark and exposed to W light for 5 min. Total soluble protein was then extracted and used for CoIP with αGFP magnetic beads. Input and eluate fractions were analysed by SDS-PAGE and immunoblotting with αHA and αmCherry (**B**) or αHA

*Figure 5 continued on next page*

**Figure 5 continued**

antibodies (D). (E) CoIP from transgenic Arabidopsis lines. Four-day-old dark-grown Arabidopsis seedlings expressing *p35S:HA-YFP-NOT9B* or *-NOT9B ΔPNB* were exposed to FR light for 5 hr followed by 5 min R (Pfr) or long-wavelength FR light (Pr). Total soluble protein was then extracted and used for CoIP with αGFP magnetic beads. Input and eluate fractions were analysed by SDS-PAGE and immunoblotting with αHA and αphyA antibodies. *, unspecific band; ▶, phyA. (F) Nuclear body formation of NOT9B and NOT9B ΔPNB. Four-day-old dark-grown Arabidopsis seedlings expressing *p35S: HA-YFP-NOT9B* (NOT9Box WT, line 6d) or *-NOT9B ΔPNB* (NOT9Box ΔPNB) were exposed to FR light for 6 hr or kept in the dark and analysed by epifluorescence microscopy. Scale bar represents 5 μm. p, p-body; ph, photobody; n, nucleus. (G) Quantification of p-bodies. Seedlings expressing *p35S:HA-YFP-NOT9B* (NOT9Box) or *p35S:HA-YFP-NOT9B ΔPNB* (NOT9B ΔPNBox) were grown for 4 days in D. Pictures were taken and number of p-bodies was counted in a 200 × 350 pixel area. Boxplots represent 12 evaluated areas per line. Letters indicate levels of significance as determined by one-way ANOVA followed by post-hoc Tukeys HSD test; $p < 0.05$. (H) Phenotype of NOT9B ΔPNBox seedlings. Seedlings were grown for 4 days in FR light (FR, 40 μmol m$^{-2}$ s$^{-1}$) or in the dark. Scale bar represents 2 mm. (I) Western blot analysis of protein levels. Total protein was extracted from 4-day-old dark-grown Arabidopsis seedlings expressing *p35S:HA-YFP-NOT9B* (NOT9Box, line 6d) or *-NOT9B ΔPNB* (NOT9B ΔPNBox) and analysed by SDS-PAGE and immunoblotting with αGFP and αACTIN antibodies. ACTIN was detected as loading control. (J) Fluence rate response curve for inhibition of hypocotyl growth. Arabidopsis seedlings expressing *p35S:HA-YFP-NOT9B* (NOT9Box, line 6d) or *-NOT9B ΔPNB* (NOT9B ΔPNBox) were grown for 4 days in FR light. Mean hypocotyl length (±SE) of 20 seedlings relative to dark-grown seedlings is shown.

NOT1 binding). To further investigate the function of the PNB site, we generated Arabidopsis NOT9B ΔPNBox lines (*p35S:HA-YFP-NOT9B ΔPNB* in Col-0). Confirming the previous finding, phyA did not co-precipitate with NOT9B ΔPNB (*Figure 5E*) and, in line with impaired phyA binding, NOT9B ΔPNB did not form phyA-dependent photobodies (*Figure 5F*). However, NOT9B ΔPNB still formed p-bodies and colocalised with DCP1, suggesting that the ΔPNB mutation does not lead to a generally misfolded protein (*Figure 5F,G*, *Figure 4—figure supplement 2*).

We then compared hypocotyl growth of NOT9Box and NOT9B ΔPNBox lines expressing the transgene at similar levels (*Figure 5H–J*). Wildtype and NOT9B ΔPNBox seedlings grown in FR light were indistinguishable, whereas NOT9Box seedlings had considerably longer hypocotyls than the wildtype and NOT9B ΔPNBox. This suggests that the PNB site, and therefore the interaction with NOT1 and/or phyA, is important for the impact of NOT9B on phyA-mediated light signalling.

## Photoactivated phyA can displace NOT9B from the CCR4-NOT complex

NOT1 and phyA both interact with NOT9B through the PNB site (*Figure 6A*), therefore we used Y3H assays to test if they compete for binding to NOT9B. In yeast cells expressing AD-NOT9B and phyA-BD, we co-expressed either NOT1 M, NOT1 MIF4G-GFP, or NOT1 DUF. Co-expression of either NOT1 M or NOT1 DUF strongly reduced growth on medium selective for interaction of NOT9B and phyA, whereas NOT1 MIF4G-GFP, which lacks the NOT9B-binding site, had no effect (*Figure 6B*). None of the NOT1 fragments competed with FHY1 for binding to phyA, suggesting that co-expression of NOT1 does not generally interfere with interaction of phyA and phyA-binding proteins (*Figure 6—figure supplement 1A*). To confirm that expression of AD-NOT9B and phyA-BD is not reduced by co-expression of NOT1 fragments, we used HA-tagged AD-NOT9B and NOT1 M, and LUC-tagged phyA-BD for Y3H competition analysis. Co-expression of HA-NOT1 M clearly reduced growth of cells expressing the Y2H pair HA-AD-NOT9B/phyA-LUC-BD but had only a minor effect on HA-AD-NOT9B and phyA-LUC-BD protein levels (*Figure 6—figure supplement 1B,C*).

To test for competition in planta, we took advantage of the findings that recruitment of NOT9B into photobodies depends on interaction with phyA (*Figures 1E,F* and *5F*) and that NOT1 does not form photobodies nor binds phyA (*Figure 6—figure supplement 2*). Thus, if there is competition for binding to NOT9B, we expect that high expression of NOT1 would hinder recruitment of NOT9B into photobodies (*Figure 6—figure supplement 3*). In tobacco leaves transiently expressing NLS-YFP-NOT9B and phyA-NLS-CFP, co-expression of a NOT1 fragment impaired in binding NOT9B (FLAG-myc-mCherry-NLS-NOT1 MIF4G-mCherry) had no effect on recruitment of NOT9B into photobodies (*Figure 6C*). In contrast, co-expression of FLAG-myc-mCherry-NLS-NOT1 M, which includes the NOT9B binding site of NOT1, almost fully abolished assembly of NOT9B into photobodies whilst not affecting phyA photobody formation. This finding is consistent with competition of phyA and NOT1 for binding NOT9B in planta. To further investigate this notion, we generated stable transgenic Arabidopsis lines expressing *p35S:FLAG-myc-mCherry-NLS-NOT1 M* either in Col-0 or NOT9Box backgrounds. FLAG-myc-mCherry-NLS-NOT1 M co-purified with HA-YFP-NOT9B in CoIPs (*Figure 6D*), which is in agreement with data shown in *Figures 3E,F* and *4C,D*. Interestingly, the

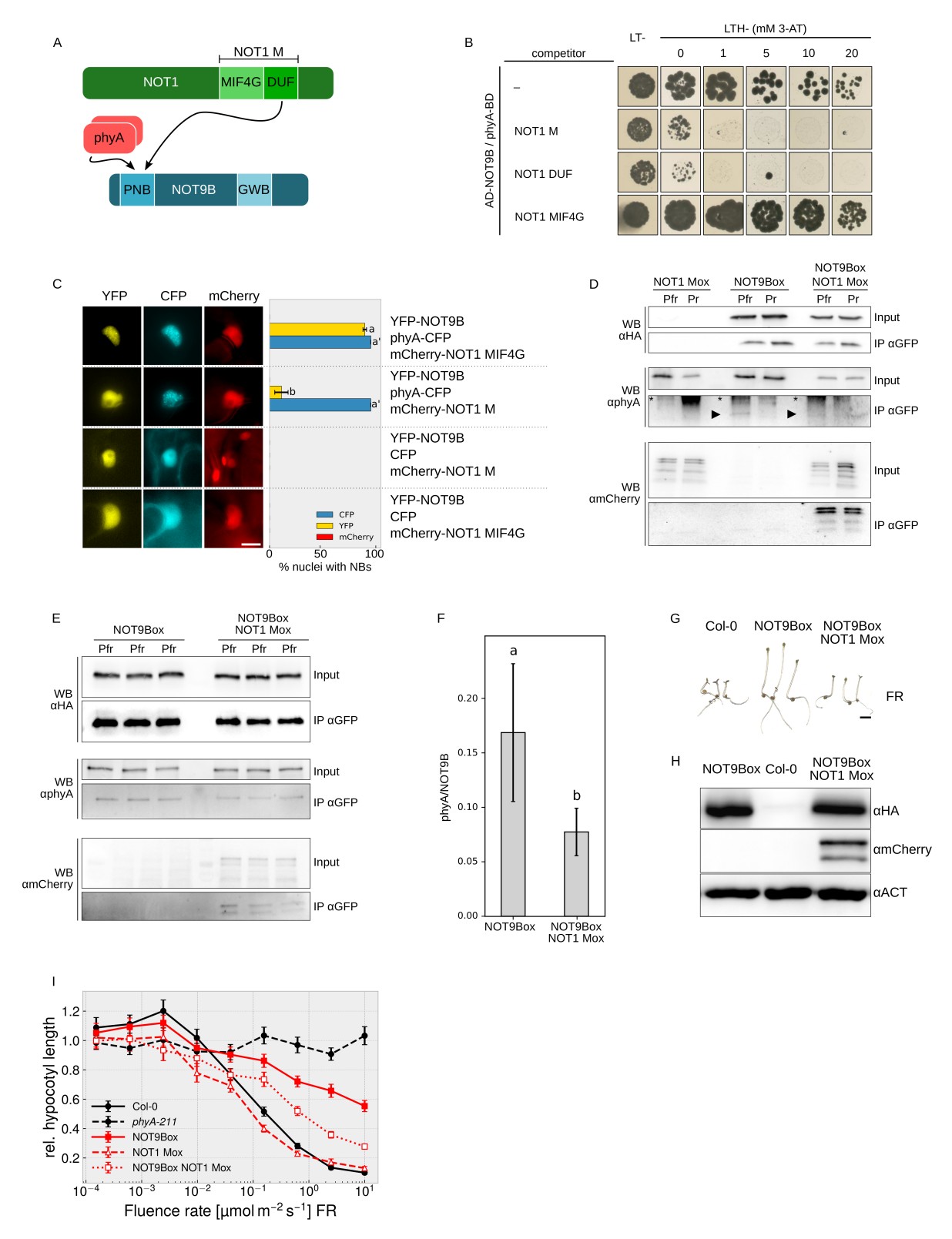

**Figure 6.** NOT1 and phyA compete for binding NOT9B. (**A**) Domain structure and interaction domains of NOT1 and NOT9B. (**B**) Yeast three-hybrid competition assay. AH109 cells were transformed with plasmids coding for the yeast-two-hybrid pair AD-NOT9B and phyA-BD, and for different NOT1 fragments as potential competitors. Yeast cells were grown in R light on CSM LTH- plates supplemented with 20 μM phycocyanobilin and containing increasing amounts of 3-AT. CSM LT- plates were used as transformation control. (**C**) Quantification of NOT9B nuclear body formation. Four-week-old

*Figure 6 continued on next page*

*Figure 6 continued*

tobacco plants were co-infiltrated with Agrobacteria transformed with plasmids coding for *p35S:NLS-YFP-NOT9B* (YFP-NOT9B), either *p35S:PHYA-NLS-CFP* (phyA-CFP) or *p35S:CFP* (CFP), and either *p35S:FLAG-myc-mCherry-NLS-NOT1 M* (mCherry-NOT1 M) or *-NOT1 MIF4G-mCherry* (mCherry-NOT1 MIF4G). Plants were kept in the dark for 2 days and subjected to epifluorescence microscopy. Representative pictures are shown on the left. Scale bar represents 5 μm. Twenty-five nuclei per plasmid combination were randomly chosen and evaluated for nuclear body formation. Bars represent the mean of three biological replicates (± SD) and show the relative number of nuclei with nuclear bodies in the respective channel. (C and F) Letters indicate levels of significance as determined by one-way ANOVA followed by post-hoc Tukeys HSD test; p<0.05. (D) CoIP from transgenic Arabidopsis lines. Four-day-old dark-grown Arabidopsis seedlings expressing *p35S:HA-YFP-NOT9B* (NOT9Box, line 21c), *p35S:FLAG-myc-mCherry-NLS-NOT1 M* (NOT1 Mox) or both were exposed to FR light for 5 hr followed by 5 min R (Pfr) or long-wavelength FR light (Pr). Total soluble protein was then used for CoIP with αGFP magnetic beads. Input and eluate fractions were analysed by SDS-PAGE and immunoblotting with αHA, αphyA, and αmCherry antibodies. *, unspecific bands; ▶, phyA. (E) CoIP for quantification. Arabidopsis NOT9Box (line 21c) and NOT9Box NOT1 Mox seedlings were grown as described in (D). CoIP and analysis of input and eluate fractions was done as in (D). The three lanes per genotype represent independent biological replicates. (F) Quantification of CoIP in (E). The amount of precipitated NOT9B and phyA was quantified using ImageJ. Bars represent mean phyA/NOT9B ratio of three biological replicates (± SD). (G) Seedling phenotype. Arabidopsis wildtype (Col-0), NOT9Box (line 21c), and NOT9Box NOT1 Mox seedlings were grown for 4 days in FR light (40 μmol m$^{-2}$ s$^{-1}$). Scale bar represents 2 mm. (H) Western blot analysis of protein levels. Total protein was extracted from 4-day-old dark-grown Col-0, NOT9Box (line 21c), and NOT9Box NOT1 Mox seedlings and analysed by SDS-PAGE and immunoblotting with αHA, αmCherry, and αACTIN antibodies. ACTIN was detected as loading control. (I) Fluence rate response curve for inhibition of hypocotyl growth. Seedlings of the indicated genotypes were grown for 4 days in FR light. Mean hypocotyl length (± SE) of 20 seedlings relative to dark-grown seedlings is shown.

The online version of this article includes the following figure supplement(s) for figure 6:

**Figure supplement 1.** Yeast three-hybrid competition assays.

**Figure supplement 2.** NOT1 does not interact with phyA and is not recruited into photobodies.

**Figure supplement 3.** Model for NOT9B photobody formation in wildtype and NOT1 Mox background.

**Figure supplement 4.** Comparison of NOT9B and phyA protein levels.

**Figure supplement 5.** NOT1 M overexpression partially rescues NOT9Box gene expression defects.

amount of phyA Pfr co-precipitating with HA-YFP-NOT9B appeared to be lower in CoIPs from seedlings co-expressing FLAG-myc-mCherry-NLS-NOT1 M than from seedlings not expressing NOT1 M (*Figure 6D*). Quantification of three independent CoIP experiments analysed on the same membrane showed that the amount of phyA bound to NOT9B is significantly reduced when NOT1 M is coexpressed (*Figure 6E,F*). This finding supports the idea that binding of phyA and NOT1 to NOT9B is mutually exclusive and they compete for binding NOT9B in planta. Therefore, increasing the levels of one of the proteins binding to the PNB site of NOT9B would decrease binding of the other protein interacting with this site. Since phyA levels in the nucleus are likely much higher and more dynamic than levels of NOT1, the impact of phyA on NOT1/NOT9B would be of greater importance in planta than the effect of NOT1 on the NOT9B/phyA interaction. Such competition-based regulation of NOT1/NOT9B interaction by phyA is favoured by phyA levels that are higher than NOT9B levels, and therefore we compared the luciferase activity in a *phyA-211 pPHYA:PHYA-LUC* complementation line and *not9b-2* complemented with *pNOT9B:LUC-NOT9B*. Quantification of the luciferase signal showed that levels of phyA are several orders of magnitude higher than the levels of NOT9B (*Figure 6—figure supplement 4*). Consequently, based on the phenotype of *not9b* mutants and NOT9Box lines, we propose that CCR4-NOT complexes containing NOT9B (CCR4-NOT[NOT9B]) repress phyA signalling. In seedlings exposed to FR light, phyA is transported into the nucleus and competition for binding NOT9B could then displace NOT9B from CCR4-NOT[NOT9B] (*Figure 6A*, *Figure 6—figure supplement 3*), thereby relieving repression of light signalling. A potential explanation for the NOT9Box phenotype is that phyA levels are not high enough to fully prevent NOT9B from binding NOT1 when NOT9B is overexpressed. If so, blocking the PNB site in NOT9B by overexpression of NOT1 M should alleviate the NOT9Box phenotype. In line with this reasoning, the negative effect of NOT9Box on both inhibition of hypocotyl growth in FR light and expression of FR light-induced genes such as *EARLY LIGHT INUDCIBLE PROTEIN 1* and *2* (*EPLI1/2*) and *CHS* (*Harari-Steinberg et al., 2001*; *Zhou et al., 2005*) is partially suppressed by co-expression of FLAG-myc-mCherry-NLS-NOT1 M without affecting NOT9B protein levels (*Figure 6G–I*, *Figure 6—figure supplement 5*).

## AP-MS analysis of NOT9B interactome

Recent AP-MS approaches to characterise the CCR4-NOT complex in plants (*Zhou et al., 2020*; *Arae et al., 2019*) and data shown in *Figure 3* support the notion that a complex similar to the CCR4-NOT complex from yeast and animals also exists in plants. We propose that phyA prevents the assembly of NOT9B into the CCR4-NOT complex, which offers a potential explanation for regulation of CCR4-NOT by FR light. A wide range of functions in control of RNA stability and metabolism have been described for yeast and animal CCR4-NOT but it is still unknown how CCR4-NOT modulates phyA mediated light responses in plants. Therefore, we investigated the NOT9B interactome in the NOT9Box line. Silver staining of GFP-trap purified proteins revealed several proteins co-precipitating with HA-YFP-NOT9B but not with YFP-HA (*Figure 7—figure supplement 1A*). By MS-MS we identified 450 proteins that are only present in the HA-YFP-NOT9B eluate fraction (*Figure 7—figure supplement 1B*, *Supplementary file 1*). This includes the known NOT9B interactors NOT1 and phyA, and most components of the CCR4-NOT complex. We did not find NOT9A, the closest homologue of NOT9B, and the more distantly related NOT9C. Interestingly, AGO1 and proteins with a function in miRNA biogenesis and/or splicing were among NOT9B-associated proteins. Highly enriched GO terms among NOT9B-associated proteins were rRNA binding, mRNA binding, RNA binding, and translation elongation factor activity (*Figure 7—figure supplement 1C*), supporting a conserved role of the CCR4-NOT complex in mRNA metabolism in eukaryotes, including plants. In the following, we investigated a potential function of AGO1 and alternative splicing in NOT9B-mediated modulation of phyA signalling.

## Nuclear localised NOT9B associates with AGO1

We did not find any evidence for direct interaction of NOT9B and AGO1 in Y2H assays (*Figure 7—figure supplement 2*) but could co-precipitate endogenous AGO1 with HA-YFP-NOT9B, and vice versa found HA-YFP-NOT9B associated with endogenous AGO1 in CoIPs with αAGO1 antibodies (*Figure 7A,B*). We also observed colocalisation of HA-YFP-NOT9B and AGO1-CFP in p-bodies of transiently transformed tobacco leaves (*Figure 4—figure supplement 1A*). Overall, AP-MS and CoIP data show that AGO1 and NOT9B are in complex but possibly do not directly interact.

AGO1 is most known for its function in miRNA-mediated cleavage of target mRNAs in the cytosol, but recent work has shown that Arabidopsis AGO1 also has a function independent of miRNAs, and binds to chromatin and regulates gene expression in the nucleus (*Bajczyk et al., 2019*; *Bologna et al., 2018*; *Liu et al., 2018*). In CoIP assays, similar amounts of NOT9B co-purified with AGO1 from lines overexpressing wildtype and constitutively nuclear localised NOT9B, suggesting that NOT9B associates with AGO1 in the nucleus (*Figure 7B*).

## AGO1 plays a role in light signalling

The function of AGO1 in regulation of gene expression in the nucleus is independent of miRNAs but requires small RNAs that define AGO1 target sites in the genome (*Liu et al., 2018*). Several genes involved in light signalling or light-regulated processes such as hypocotyl growth, anthocyanin biosynthesis, establishment of photosynthetic capacity, photoperiodic flowering, or circadian rhythms are among AGO1-bound genes, suggesting that the nuclear localised fraction of AGO1 might be involved in modulation of light responses (*Liu et al., 2018*). *Sorin et al., 2005* have shown that hypocotyl growth at intermediate light intensities is reduced in weak, viable *ago1* mutants (e.g. *ago1-33*, *ago1-34*, and *ago1-35*) compared to the wildtype. To further investigate a potential function of AGO1 in light signalling, we measured detailed fluence rate response curves in FR light for the weak *ago1-27* allele, and quantified anthocyanin levels and cotyledon opening. The *ago1-27* mutant has only slightly shorter hypocotyls than the wildtype in the dark but is hypersensitive to FR light over a wide range of fluence rates (*Figure 7C*, *Figure 7—figure supplement 3*). In addition anthocyanin levels are increased in *ago1-27* exposed to FR light, whereas there is no significant difference in the dark and in B light (*Figure 7D*). Enhanced accumulation of anthocyanin in FR light is therefore wavelength-specific and not a general light or stress phenotype. Cotyledon unfolding is typically enhanced by light but we found that this response is partially impaired in the *ago1-27* and the *not9b-1* mutants (*Figure 7E,F*). Thus, while generally being hypersensitive to FR light, *ago1-27* and *not9b-1* are hyposensitive regarding cotyledon unfolding, which distinguishes them from other light signalling mutants and may indicate that NOT9B and AGO1 are functionally linked. Interestingly, the

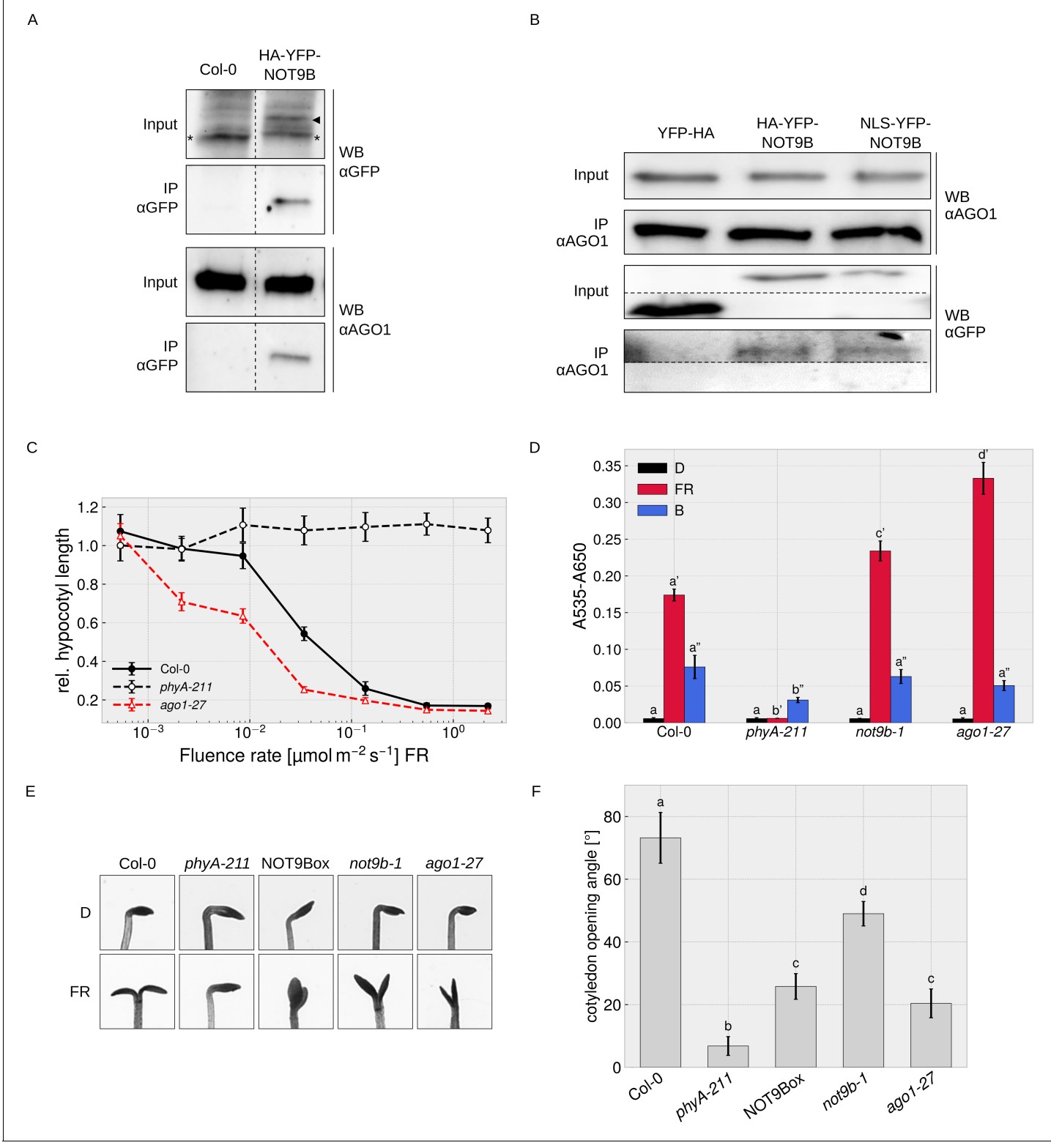

**Figure 7.** Physical and phenotypical link between NOT9B and AGO1. (**A**) CoIP from transgenic Arabidopsis lines. Total soluble protein was extracted from 4-day-old dark-grown wildtype (Col-0) and *p35S:HA-YFP-NOT9B* expressing seedlings (NOT9Box) and used for CoIP with αGFP magnetic beads. Input and eluate fractions were analysed by SDS-PAGE and immunoblotting with αGFP and αAGO1. *, unspecific band; ▶, HA-YFP-NOT9B. (**B**) CoIP from Arabidopsis lines expressing constitutively nuclear localised NOT9B. Total soluble protein was extracted from 4-day-old dark-grown Arabidopsis seedlings expressing *p35S:YFP-HA* (YFP-HA), *p35S:HA-YFP-NOT9B* (NOT9Box), or *p35S:NLS-YFP-NOT9B* (NLS-YFP-NOT9B) and used for CoIP with

*Figure 7 continued on next page*

*Figure 7 continued*

αAGO1 bound to protein A-coated magnetic beads. Input and eluate fractions were analysed as in (**A**) using αAGO1 and αGFP antibodies. (**C**) Fluence rate response curve for inhibition of hypocotyl growth. Seedlings of the indicated genotypes were grown for 4 days in FR light. Mean hypocotyl length (± SE) of 20 seedlings relative to dark-grown seedlings is shown. Absolute hypocotyl length is shown in **Figure 7—figure supplement 3**. (**D**) Quantification of anthocyanin content. Seedlings of indicated genotypes were grown on ½ MS supplemented with 1.5% sucrose for 4 days in the dark (D), FR, or B light. Anthocyanin was extracted from 25 seedlings per genotype/condition and $A_{535}$-$A_{650}$ was measured. Bars represent biological triplicates (± SD). (**D** and **F**) Letters indicate levels of significance as determined by one-way ANOVA followed by post-hoc Tukeys HSD test; $p<0.05$. (**E** and **F**) Cotyledon opening in FR light. Seedlings of indicated genotypes were grown for 4 days in D followed by 24 hr in weak FR light (716 nm, 0.05 $\mu$mol m$^{-2}$ s$^{-1}$) or D. (**F**) Cotyledon opening of FR-treated seedlings was quantified by measuring the angle between the cotyledons of 20 seedlings per genotype. Bars show mean cotyledon opening of three biological replicates (± SD).

The online version of this article includes the following figure supplement(s) for figure 7:

**Figure supplement 1.** Investigation of the NOT9B interactome by AP-MS.
**Figure supplement 2.** Yeast-two-hybrid growth assay for NOT9B and AGO1.
**Figure supplement 3.** Absolute hypocotyl length of *ago1-27* seedlings.

NOT9Box line is also impaired in proper cotyledon unfolding in FR light (**Figure 7E,F**), for which we present a potential explanation in the discussion.

## NOT9B affects alternative splicing

Since the NOT9B interactome includes proteins with a function in splicing (**Figure 7—figure supplement 1**), we searched the literature for reports on FR light regulated splicing events. MYBD is a MYB-related transcription factor that promotes anthocyanin biosynthesis downstream of HY5 (**Nguyen et al., 2015**). *MYBD* is regulated by alternative splicing; splice variant *MYBD.1* encodes a functional protein, while *MYBD.2* has a premature stop codon due to intron retention (**Figure 8A**; **Hartmann et al., 2016**). The ratio of *MYBD.2/.1* is strongly reduced upon R or FR light treatment, meaning that light shifts the *MYBD.2/.1* ratio toward the splice variant that codes for functional MYBD and thereby promotes anthocyanin biosynthesis (**Hartmann et al., 2016**; **Nguyen et al., 2015**). We found that phyA is required for this response to FR light (**Figure 8B**) and that the *MYBD.2/.1* ratio in dark-grown *not9b-2* seedlings is similar to wildtype seedlings treated with FR light (**Figure 8C**). Thus, NOT9B is essential to suppress the shift of the *MYBD.2/.1* ratio toward the physiologically active splice variant in etiolated seedlings and FR light perceived by phyA removes this negative regulation.

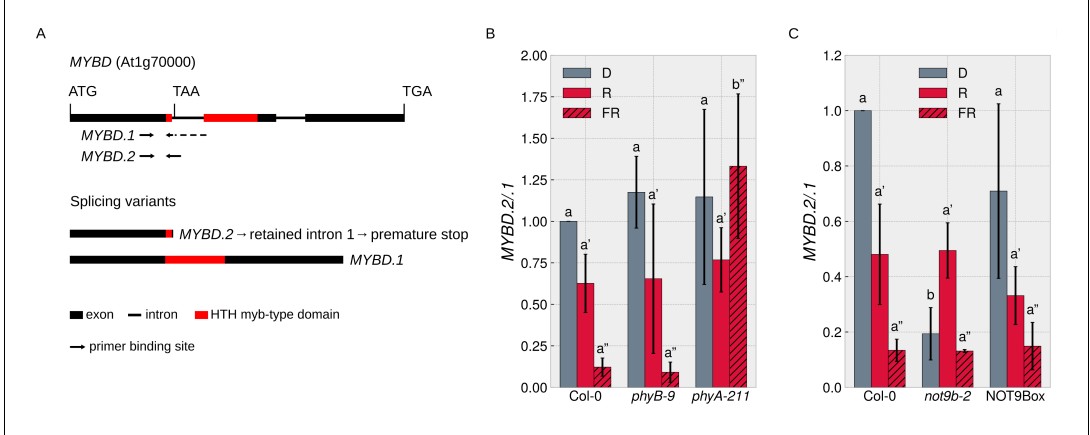

**Figure 8.** NOT9B affects alternative splicing of *MYBD*. (**A**) Schematic overview of *MYBD* genomic locus and *MYBD* transcripts. Alternative splicing of *MYBD* results in two transcripts, *MYBD.1* and *MYBD.2*. Binding sites of splicing variant specific primers used in (**B** and **C**) are indicated. (**B** and **C**) Analysis of *MYBD* splicing pattern. Six-day-old dark-grown seedlings of indicated genotypes were exposed to R or FR light for 6 hr or kept in the dark. Total RNA was extracted and transcribed into cDNA. Splice site specific primers and qPCR were used to quantify the ratio between the two splice variants (*MYBD.2/MYBD.1*). Bars show mean *MYBD.2/MYBD.1* ratio of three biological replicates (± SD). Letters indicate levels of significance as determined by one-way ANOVA followed by post-hoc Tukeys HSD test; $p<0.05$.

## The *not9b* mutant shows a partial *cop* phenotype

Seedlings lacking functional *NOT9B* do not have an obvious *constitutively photomorphogenic* (*cop*) phenotype (*Figures 2D* and *7E*, *Figure 2—figure supplement 3A*), but the splicing pattern of *MYBD* in etiolated *not9-2* resembles the pattern in wildtype seedlings exposed to light (*Figure 8C*). Thus, as an additional molecular read out of light signalling, we quantified transcript levels of the early light response genes *EARLY PHYTOCHROME RESPONSIVE 1* (*EPR1*)/*REVEILLE 7* (*RVE7*), *ELIP1*, and *ELIP2* (*Kuno, 2003*; *Harari-Steinberg et al., 2001*). Expression of *EPR1*, *ELIP1*, and *ELIP2* was increased in dark-grown *not9b-2* seedlings compared to the wildtype, supporting the concept that *not9b* mutant seedlings have a partial *cop* phenotype at the molecular level (*Figure 9A*). When exposed to FR light for 1 hr, *EPR1*, *ELIP1*, and *ELIP2* transcript levels are similar in *not9b-2* and wild-type seedlings. Expression of *EPR1* and *ELIP1* was also increased in dark-grown NOT9Box seedlings, although not to the same extent as in *not9b-2*.

## Discussion

Seedlings with altered NOT9B levels have a FR light-specific phenotype and do not show any obvious developmental defects despite the evolutionary conservation and general function of the CCR4-NOT complex (*Figure 2*, *Figure 2—figure supplement 1*). This is in stark contrast to many other mutants with defects in complexes or pathways of general function, such as splicing and miRNA biogenesis. Several of these mutants show altered responses to light but they are not wavelength-specific, and often these mutants have highly pleiotropic phenotypes with loss-of-function alleles being lethal (*Xin et al., 2019*; *Xin et al., 2017*; *Shikata et al., 2012*; *Laubinger et al., 2008*; *Sorin et al., 2005*). While *CAF40*/*CNOT9* are single copy genes in humans and yeast, plants generally contain two or more genes coding for proteins with similarity to CAF40/CNOT9, including *NOT9A*, *NOT9B*, and *NOT9C* in Arabidopsis (*Zhou et al., 2020*; *Arae et al., 2019*). Residues required for recruitment into p-bodies are conserved in all NOT9 proteins, including the more distantly related NOT9C, and consistently, we found that all Arabidopsis NOT9 proteins form p-bodies (*Figure 1—figure supplements 1* and *3C*). The region corresponding to the PNB site of NOT9B is more variable (*Figure 1—figure supplement 1*). NOT9C is most divergent and did not interact with phyA and NOT1 nor did it co-purify with CCR4-NOT in previous AP-MS approaches (*Figure 1—figure supplement 3A,B*; *Zhou et al., 2020*; *Arae et al., 2019*). In contrast, NOT9A and NOT9B both bind NOT1 but we only observed interaction of phyA and NOT9B (*Figure 1*, *Figure 1—figure supplement 3A,B*). We hypothesise that residues in NOT9A and NOT9B that correspond to the residues mutated in NOT9B ΔPNB possibly contribute to specify the NOT9A/B interaction profile. Y2H and CoIP from HEK293T cells show that no additional plant-specific proteins are required for NOT9B/phyA complex formation but we cannot formally rule out the possibility that components of the CCR4-NOT complex conserved in animals, yeast, and plants bridge between NOT9B and phyA in these assays. Yet, if phyA would bind indirectly to NOT9B through other components of the CCR4-NOT complex, we would also expect complex formation for phyA and both NOT9A and NOT1 in the Y2H assay and recruitment of NOT9A and NOT1 into phyA-dependent photobodies. However, we did not observe either (*Figure 1—figure supplement 3A,B,C*, *Figure 6—figure supplement 2*) and therefore favour a model in which phyA directly binds NOT9B.

NOT9A and NOT9B co-precipitated with NOT1, NOT3, and CCR4b in previous AP-MS approaches (*Zhou et al., 2020*; *Arae et al., 2019*) but we did not find NOT9A among NOT9B associated proteins in our AP-MS experiment (*Figure 7—figure supplement 1*). Thus, two pools of CCR4-NOT complexes, CCR4-NOT$^{NOT9A}$ and CCR4-NOT$^{NOT9B}$, may exist that contain either NOT9A or NOT9B. Whether they differ regarding functionality is still unknown, but we expect that CCR4-NOT$^{NOT9A}$ and CCR4-NOT$^{NOT9B}$ are different in terms of regulation by light, since phyA binds NOT9B but not NOT9A.

PhyA and NOT1 bind to the PNB site of NOT9B and several approaches provide evidence that binding to NOT9B is mutually exclusive (*Figure 6*). Increasing the levels of one PNB binding protein will therefore reduce interaction of NOT9B with the other PNB binding protein. PhyA is several orders of magnitude more abundant than NOT9B (*Figure 6—figure supplement 4A*), and it is also expected to be more abundant and more dynamic than NOT1. Under such conditions, phyA likely has a greater impact on the regulation of the NOT9B-NOT1 interaction, than NOT1 on the phyA-NOT9B interaction. Nuclear localised NOT9B is sufficient for modulation of phyA mediated light

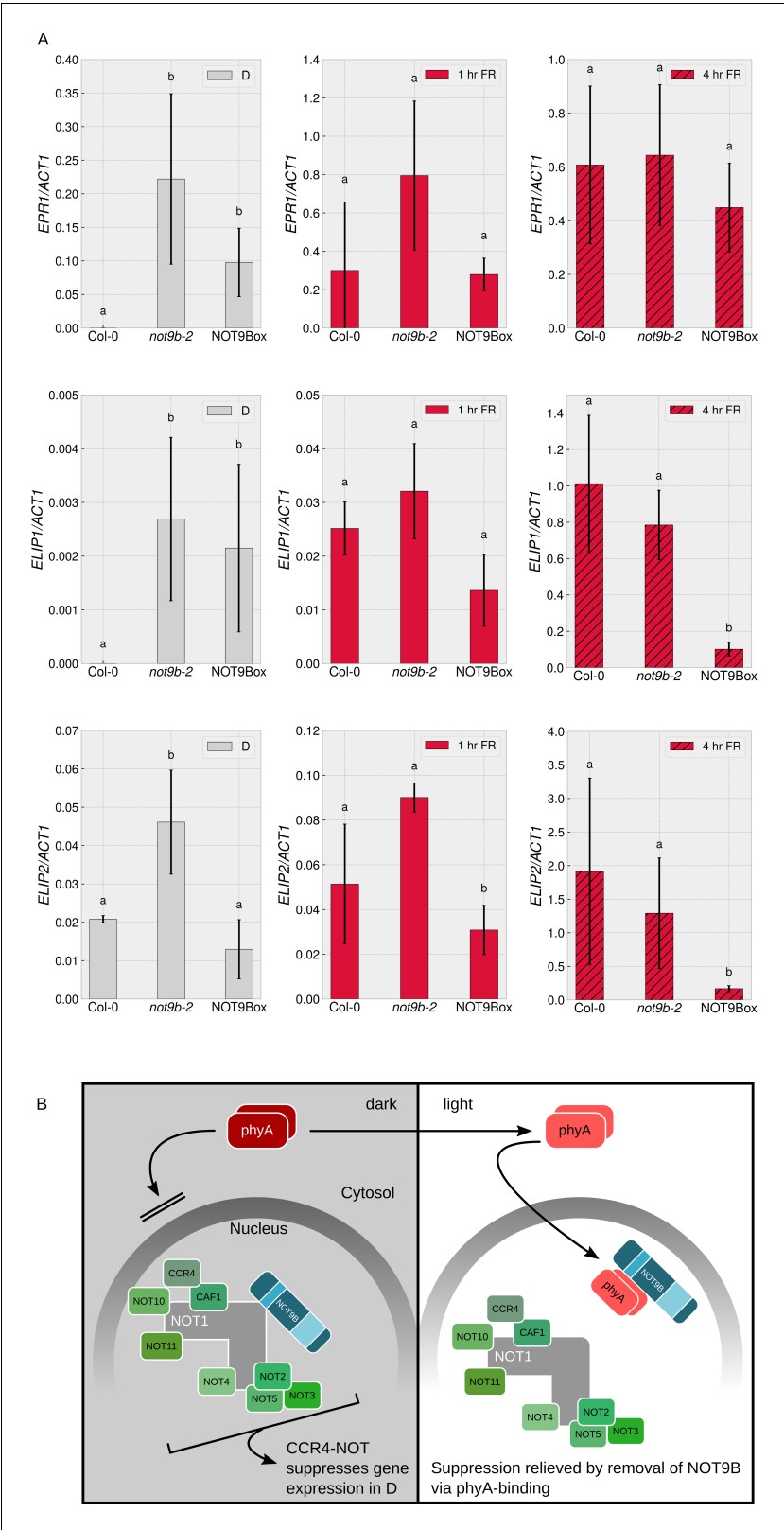

**Figure 9.** NOT9B suppresses gene expression in darkness. (**A**) Quantification of *EPR1*, *ELIP1*, and *ELIP2* transcript levels in wildtype (Col-0), *not9b-2*, and NOT9Box seedlings. Four-day-old dark-grown seedlings were exposed to FR light for 1 or 4 hr or kept in the dark. Total RNA was extracted and *EPR1*, *ELIP1*, *ELIP2*, and *ACT1* transcript levels were quantified by qRT-PCR. Bars represent biological triplicates (± SD). Letters indicate levels of significance as determined by one-way ANOVA followed by post-hoc Tukeys HSD test; p<0.05. (**B**) Hypothetical working model for modulation of phyA-

*Figure 9 continued on next page*

*Figure 9 continued*

mediated light signalling by nuclear NOT9B. The CCR4-NOT^NOT9B complex suppresses gene expression in the dark. Light-induced activation and translocation into the nucleus allow phyA to bind NOT9B and displace it from CCR4-NOT^NOT9B, thereby promoting expression of light-regulated genes.

The online version of this article includes the following figure supplement(s) for figure 9:

**Figure supplement 1.** Hypothetical model of NOT9B in a ternary complex.

signalling, which therefore does not depend on p-bodies (*Figure 4I–L*). We propose that upon transport into the nucleus, phyA binds NOT9B and thereby displaces NOT9B from the CCR4-NOT^NOT9B complex (*Figure 9B*). Based on physiological and gene expression data, we propose a NOT1/phyA competition model and assume that CCR4-NOT^NOT9B is the active unit that represses light signalling while the complex without NOT9B does not affect light responses. Both the light-dependence of phyA nuclear transport and the preferential interaction of NOT9B with phyA in the active Pfr state would contribute to disassembly of CCR4-NOT^NOT9B specifically in FR light, which could explain the phyA- and FR light-specific phenotype of *not9b* mutant and NOT9Box seedlings. In an alternative phyA repression model, NOT9B could bind phyA and repress phyA activity. In contrast to the NOT1/phyA competition model, the effect of NOT9B on light signalling would be independent of the CCR4-NOT complex in the phyA repression model. However, we favour the NOT1/phyA competition model over the phyA repression model for three reasons. First of all, levels of endogenous NOT9B are much lower than levels of phyA (*Figure 6—figure supplement 4A*). Under such conditions, efficient competition between NOT1 and phyA for binding NOT9B is possible for which we have experimental evidence (*Figure 6*). In contrast, it appears unlikely that NOT9B levels, that are orders of magnitude lower than phyA levels, can efficiently repress phyA activity. Further insight could be gained by affinity measurements using heterologously expressed proteins, which also could support the notion that the phyA-NOT9B interaction is direct. Secondly, dark-grown *not9b* mutant seedlings show a partial *constitutively photomorphogenic* (*cop*) phenotype and are unable to fully suppress photomorphogenesis in the dark (*Figures 8* and *9A*). This finding is consistent with the NOT1/phyA competition model, whereas dark-grown *not9b* seedling should be fully etiolated in the phyA repression model. Lastly, if the phyA repression model were true, we would expect all phyA-regulated responses to be increased in the *not9b* mutant, which is not the case (i.e. cotyledon unfolding in FR light is reduced in *not9b*).

In general, *not9b* mutant and NOT9Box seedlings have opposite phenotypes. However, we observed that both *not9b* and NOT9Box seedlings are hyposensitive to light with regard to cotyledon opening and both show increased expression of a subset of light-induced genes in the dark (*Figures 7E,F* and *9A*). A possible explanation for this seemingly counterintuitive observation is that NOT9B is part of a multimeric complex and could act as a linker protein between CCR4-NOT^NOT9B and other proteins bound to NOT9B. In contrast to a dimeric complex, where increasing the level of one component does not have a negative effect on complex formation, there is an optimal concentration of the bridge protein at which a ternary complex is most abundant (*Figure 9—figure supplement 1A,B*; *Douglass et al., 2013*). For a hypothetical NOT1-NOT9B-protein X ternary complex, in which NOT9B is the bridge protein, increasing NOT9B levels above the optimal concentration would predominantly result in formation of NOT1-NOT9B and NOT9B-protein X dimeric complexes at the expense of the NOT1-NOT9B-protein X ternary complex (*Figure 9—figure supplement 1B–E*). Thus, if only the ternary complex is functional, *not9b* and NOT9Box seedlings can have the same phenotype. Different protein Xs may exist (protein $X_A$ and $X_B$ in *Figure 9—figure supplement 1F*) and associate with NOT9B to regulate light responses in a context-specific manner. The optimal concentration of NOT9B leading to highest levels of NOT1-NOT9B-protein X complexes depends on the concentration of protein $X_A$ and $X_B$ and their affinity for NOT9B. Therefore, overexpression of NOT9B may increase NOT9B levels to concentrations that favour the formation of ternary complexes with one protein X (protein $X_B$ in *Figure 9—figure supplement 1F*), while the same NOT9B levels may exceed this optimal concentration for the assembly of ternary complexes with another protein X (protein $X_A$ in *Figure 9—figure supplement 1F*). As a consequence, for some responses, NOT9Box lines and *not9b* mutants may show the same phenotype (responses depending on NOT1-NOT9B-protein $X_A$ in *Figure 9—figure supplement 1F*), while the phenotype of the mutant and the

overexpression line may be opposite for other responses (responses depending on NOT1-NOT9B-protein $X_B$, *Figure 9—figure supplement 1F*). However, this hypothesis is difficult to test at the moment for three reasons: firstly, the concentrations of NOT1, NOT9B, and protein Xs in planta are unknown and they may vary depending on tissue and developmental stage, secondly, binding affinities for NOT1/NOT9B and NOT9B/protein Xs are unknown, and lastly, protein Xs themselves are unknown.

A potential approach to identify candidates for protein X is to use a line expressing exclusively nuclear localised NOT9B ΔPNB for AP-MS. In such an approach, we would expect to find nuclear localised proteins that associate with NOT9B through binding sites other than the PNB site. Potential candidates for X might directly or indirectly interact with NOT9B through the GWB motif but binding to yet unknown sites is also possible.

We found that AGO1 associates, possibly indirectly, with NOT9B and therefore AGO1 could be part of a protein complex that comprises a protein X. The GWB site in NOT9B corresponds to the binding site of TNRC6/GW182 proteins in the human NOT9B homologue CNOT9 (*Chen et al., 2014*; *Mathys et al., 2014*), and AGO proteins bind to GW-repeats in TNRC6/GW182 (*Lazzaretti et al., 2009*). Plants lack sequence homologues of TNRC6/GW182 but they contain proteins with GW-repeats (*Pfaff et al., 2013*; *Karlowski et al., 2010*). Therefore, we speculate that a still unknown GW-repeat containing protein could bind NOT9B at the GWB site and link it to AGO1. A potential function of AGO1 in a CCR4-NOT[NOT9B]-AGO1 complex could be to recruit CCR4-NOT[NOT9B] to specific sites in the genome. Target sites of such complexes could be defined by small RNAs bound by AGO1 (*Liu et al., 2018*), while activities linked to CCR4-NOT could affect the transcription at the target site. Potential small RNAs in complex with CCR4-NOT[NOT9B]-AGO1 could be identified by immunoprecipitation on an NLS-YFP-NOT9B expressing line followed by smallRNAseq.

Interestingly, we found that phyA-dependent regulation of splice site selection is disturbed in the *not9b* mutant (*Figure 8*) and several proteins involved in splicing co-purified with NOT9B in the AP-MS approach (*Figure 7—figure supplement 1*). Ago2, an AGO protein in *Drosophila* was shown to regulate alternative splicing patterns (*Lee and Rio, 2015*; *Taliaferro et al., 2013*). We speculate that nuclear localised AGO1 in complex with CCR4-NOT[NOT9B] could play a role in splice site selection in Arabidopsis and that NOT9B through its specific interaction with phyA could put this event under FR light control.

In this study, we identified NOT9B as novel phyA interacting protein providing a potential link between light signalling and the evolutionary conserved CCR4-NOT complex. We show that NOT9B is involved in light-dependent development of plants, and suggest a competition-based mode of action that depends on displacement of NOT9B from the CCR4-NOT complex by light-activated phyA. These findings can provide a basis for understanding of how light signalling in plants could feed into central processes common to all eukaryotes, such as mRNA metabolism. Future research will need to reveal the molecular mechanisms that connect the evolutionary ancient CCR4-NOT complex to gene expression in plants.

# Materials and methods

**Key resources table**

| Reagent type (species) or resource | Designation | Source or reference | Identifiers | Additional information |
|---|---|---|---|---|
| Gene (*Arabidopsis thaliana*) | *NOT9B* | https://www.arabidopsis.org/ | AT5G12980 | n/a |
| Gene (*Arabidopsis thaliana*) | *PHYA* | https://www.arabidopsis.org/ | AT1G09570 | n/a |
| Gene (*Arabidopsis thaliana*) | *NOT1* | https://www.arabidopsis.org/ | AT1G02080 | n/a |

*Continued on next page*

*Continued*

| Reagent type (species) or resource | Designation | Source or reference | Identifiers | Additional information |
|---|---|---|---|---|
| Gene (*Arabidopsis thaliana*) | *AGO1* | https://www.arabidopsis.org/ | AT1G48410 | n/a |
| Cell line (*H. sapiens*) | HEK293T | https://www.atcc.org/ | 293T (ATCC CRL-3216) | Cell line maintained by Core Facility Signalling Factory, University of Freiburg, |
| Genetic reagent (*Arabidopsis thaliana*) | *not9b-1* | https://www.arabidopsis.org/, *Sessions et al., 2002* | SAIL_584_D02/NASC N824896 | |
| Genetic reagent (*Arabidopsis thaliana*) | *not9b-2* | https://www.arabidopsis.org/, *Alonso, 2003* | SALKseq_39334/NASC N925357 | |
| Genetic reagent (*Arabidopsis thaliana*) | *ago1-27* | https://www.arabidopsis.org/, *Morel et al., 2002* | n/a | |
| Genetic reagent (*Arabidopsis thaliana*) | 35S:HA-YFP-NOT9B | this paper | ATPS06 | n/a |
| Antibody | αPHYA (rabbit polyclonal) | Agrisera, Vännäs, Sweden | AS07 220 | WB (1:1,500) |
| Antibody | αAGO1 (rabbit polyclonal) | Agrisera, Vännäs, Sweden | AS09 527 | WB (1:2,000) |
| Recombinant DNA reagent | 35S:HA-YFP-NOT9B | this paper | DS361 | n/a |
| Commercial assay or kit | Anti-GFP MicroBeads | Miltenyi | 130-091-125 | n/a |
| Chemical compound, drug | Phycocyano-billin (PCB) | Frontier Scientific, Logan, Utah | FSIP14137 | n/a |

## Cloning

All vectors, oligonucleotides, and gBlocks used in this study have either been described previously or are listed in *Supplementary file 2*. All plasmid constructs have been verified by sequencing and analytical digest.

## Co-immunoprecipitation from plants

One gram of indicated plant material (infiltratred tobacco leaves or Arabidopsis seedlings) was collected under indicated light conditions and ground for 5 min in liquid $N_2$. Working in safe green light conditions at 4°C, protein was extracted using 4 ml IP buffer (pH 7.8; 134 mM $Na_2HPO_4$, 1.56 mM $NaH_2PO_4$, 450 mM NaCl, 1 mM KCl, 1 mM EDTA, 1% PEG 4000, 0.5% Triton X-100, 1 mM $Na_3VO_4$, 2 mM $Na_4P_2O_7$, 10 mM NaF), one vial Protease Inhibitor Cocktail (Sigma-Aldrich, Cat-No: I3911) per litre, 1× Protease cOmplete Inhibitor Cocktail (Sigma-Aldrich, Cat-No: 04693159001). Buffer was supplemented with 5 mM DTT if *N. benthamiana* was used as an expression system. Lysate was further homogenised using Potter-Elvehjem homogenisers. Lysate was cleared by centrifugation, the soluble fraction was separated, and 50 µl Anti-GFP MicroBeads (Miltenyi, Cat-No: 130-091-125) or Protein A beads (Miltenyi, Cat-No: 130-071-001) and indicated antibody were added and incubated under gentle shaking for 2 hr at 4°C. Columns were equilibrated and washed post pulldown according to manufacturers instructions. Elution was performed using preheated (95°C) Elution Buffer (100 mM Tris/HCl, pH 6.8, 4% SDS, 20% glycerol, 0.05% Bromphenol blue). Elution fractions were analysed by SDS-PAGE and immunoblotting.

### Co-immunoprecipitation from mammalian cell culture

HEK293T cells were used as platform for heterologous expression of proteins for co-immunoprecipitation. Culture conditions and IP conditions have been described previously (*Enderle et al., 2017*).

### Cell lines

The HEK293T cell line was provided by the Core Facility Signalling Factory, University of Freiburg; the cell line was tested negatively for mycoplasma contamination and its identity was confirmed by STR analysis.

### Plant material

All plant lines used in this study and primers used for genotyping are listed in *Supplementary file 2*.

### Seed sterilisation

Seeds were surface sterilised by incubation in 1 ml of 70% ethanol (v/v) for 10 min under constant shaking followed by washing with 1 ml 100% ethanol for 10 min under constant shaking. Seeds were dried on filter paper under sterile condition.

### Plant growth conditions

If not indicated otherwise, *Arabidopsis thaliana* seedlings were grown on four layers of filter paper (Macherey-Nagel; Cat-No: MN 615) wetted with 4.5 ml sterile $H_2O$. Seeds were stratified for 2–3 days in darkness at 4°C, germination was induced by incubation in white light (70 µmol $m^{-2}$ $s^{-1}$) for 8 hr at 22°C. Plates were afterwards kept for additional 16 hr in D at 22°C, prior to transfer to respective light conditions.

For propagation and breeding purposes, plants were grown under long day (LD) conditions (16 hr W, 70 µmol $m^{-2}$ $s^{-1}$, 22°C/8 hr D, 18°C) on standard soil (Einheitserde, Cat-No: 540203).

*N. benthamiana* plants were grown under LD conditions (16 hr W, 70 µmol $m^{-2}$ $s^{-1}$, 22°C/8 hr D, 18°C) on standard soil (Einheitserde, Cat-No: 540203).

### Measurement of hypocotyl length

Plants were grown for 4 days under the respective light conditions. 20 individual seedlings of each genotype were measured using the ImageJ software. Either the relative hypocotyl length (mean hypocotyl length in light divided by mean hypocotyl length in D) or absolute hypocotyl length is shown.

### FRET-FLIM analysis

Transient expression of fluorescently tagged (YFP and mCherry) proteins in leek epidermal cells was performed by particle bombardment using a particle gun, PDS1000/-He Biolistic Particle Delivery system (Bio-Rad, USA). 400 ng of each plasmid were mixed and added to 5 µl gold particles (1.0 µm diameter #1652263, Bio-Rad), 10 µl 2.5 M $CaCl_2$, and 4 µl 0.1 M Spermidine (S2626-1G, Sigma-Aldrich) in microcentrifuge tubes. The mixtures were then incubated at room temperature for 15 min with occasional vortexing, pulse-centrifuged (10,000 rpm for 10 s), washed initially with 100 µl 70% ethanol, followed by 50 µl 100% ethanol. The supernatants between the washes were removed by pipetting and the pellets were resuspended finally in 12 µl 100% ethanol. Macrocarriers (#165–2258, Bio-Rad) were assembled on the metal rings and the resuspended gold particles carrying the plasmids (microcarriers) were applied onto the macrocarriers. The assemblies were incubated at room temperature for 5–10 min to evaporate the ethanol, and to make the microcarriers stick to the macrocarriers. Leek pieces were prepared by separating the 3rd, 4th, and 5th layers from outside, cut into 3 × 1 inch pieces and placed in petri dishes moistened with wet paper towels. The metal rings with the macrocarriers were loaded into the particle gun and high-pressure helium gas was used to bombard the microcarriers through 900 psi rupture disks (#165–2257, Bio-Rad) in vacuum (25 inHg) through stopping screens onto the leek pieces. After 14–16 hr of incubation in darkness, the epidermal layers of bombarded leek pieces were subjected to confocal microscopy followed by FRET-FLIM (Förster Resonance Energy Transfer by Fluorescence Lifetime Imaging).

The leek cells were imaged with a Confocal Laser Scanning Microscope (Leica CLSM SP8, Leica Microsystems). For FRET-FLIM analyses, the lifetime of the donor fluorophore YFP was compared

between the leek cells that express YFP-tagged protein alone or co-expressed with mCherry-tagged proteins. YFP was excited with a pulsed picosecond laser at 470 nm (detection gate of 521–549 nm, and a frequency of 40 MHz) and the emission signals were detected by HyD SMD hybrid detector (Leica Microsystems). In the same cell the expression of mCherry- (excited by a 561 nm PMT laser with detection gating of 620–730 nm) tagged proteins (if co-expressed) was also detected. The fluorescence lifetime of YFP was determined by Time Correlated Single Photon Counting (TCSPC) using FLIM hardware from PicoQuant (PicoQuant GmbH, Berlin, Germany). The detections were limited to 1000 photons/pixel. In cells, in which the tagged proteins formed nuclear bodies (NBs), appropriate regions of interest (ROIs) were chosen for the calculation of time correlated histograms. The ROIs were expanded to nuclear peripheries in the cells that did not form any NBs. The time correlated histograms were then deconvoluted and fitted by n-exponential tailfit algorithm into mono- (for the cells expressing YFP-tagged proteins only) or bi-exponential decay (for the cells expressing YFP and mCherry-tagged proteins) curves using the FLIM package of SymphoTime 64 software (PicoQuant GmbH, Berlin, Germany). Only lifetime values from cells with $\chi^2 < 1$ were considered for analysis. The experiments were conducted on different days with independent bombardments and the data from several experiments were combined for the final calculations.

## Measurement of germination rate

Seeds were sown on ½ MS/2% agar plates and incubated in D for 2 hr. Then, a 5 min FR pulse (740 nm, 40 µmol m$^{-2}$ s$^{-1}$) was applied followed by 48 hr incubation in D. In the following, the plates were either incubated for 3 min in FR (740 nm, 40 µmol m$^{-2}$ s$^{-1}$) or for 30 s in R (660 nm, 10 µmol m$^{-2}$ s$^{-1}$) followed by incubation in D for another 5 days. One set of plates was kept in W (70 µmol m$^{-2}$ s$^{-1}$ PAR) to estimate seed viability, one set was not illuminated and kept in D. After 5 days seedlings were counted and germination frequency calculated.

## Quantification of transcript levels

Seedlings were grown for 4 days under indicated conditions. Total RNA was extracted using the Plant concert reagent (Invitrogen, Carlsbad, California) followed by RNA clean up using the RNAII plant RNA kit (Bioline, London, UK). RNA quantity and quality was assessed by UV-Vis spectroscopy.

Total RNA (1 µg) was reverse transcribed into cDNA using the High Capacity Reverse transcription kit (Thermo Fisher Scientific, Waltham, Massachusetts). qRT-PCR was performed using target-specific primer and/or probe sets as described in *Supplementary file 3*. Primers were designed exon-exon spanning, if possible. Primer and probe specificity was analysed using Primer Blast (*Ye et al., 2012*).

qPCR reactions were set up as indicated in the 2× qPCRBIO SyGreen Mix Separate-ROX manual (Nippon Genetics Europe GmbH, Düren, Germany, Cat.: PB20.14–51). 5 ng of reverse transcribed RNA was used as template, 400 nM of each primer (and probe if applicable) was used for transcript specific amplification. Reactions were set up in 384 FastGene plates (Nippon Genetics Europe GmbH, Düren, Germany, Cat.: FG-300150) and measured on a CFX384 Touch real-Time PCR Detection System (Bio-Rad Laboratories, Inc, Hercules, California). The cycling conditions were set to 2 min at 95°C, followed by 40 cycles of 5 s denaturation at 95°C and 30 s annealing/extension at 60°C. A melting curve analysis was performed for each qPCR run to identify secondary products. For each primer pair a dilution series was assayed to analyse amplification efficiency, which was used for calibration of the quantification of the starting quantity (*Livak, 1997*). Each biological replicate was measured in technical triplicates. For light treatment, plants were kept in FR (740 nm, 40 µmol m$^{-2}$ s$^{-1}$), B (30 µmol m$^{-2}$ s$^{-1}$ PAR), R (660 nm, 20 µmol m$^{-2}$ s$^{-1}$), or W light (70 µmol m$^{-2}$ s$^{-1}$ PAR) for the indicated time.

## Yeast-2-Hybrid methods

The initial Yeast-2-Hybrid screen, in which NOT9B was identified as phyA-interacting protein, was done as previously described (*Sheerin et al., 2015*).

For Yeast-2-Hybrid experiments performed in this study, yeast strain AH109 (Takara Clontech, Kyoto, Japan) was transformed with plasmids coding for AD and either BD or BD-Aux using the Frozen-EZ yeast Transformation Kit (Zymo Research, Freiburg, Germany Cat-No: T2001). Transformants were isolated on Leucin-Tryptophane dropout medium (CSM LT-), resuspended in sterile ddH$_2$O and

diluted to an $OD_{600}$ of 0.1. 5 µl of yeast suspension was spotted onto Leucin-Tryptophane-Histidine dropout medium (CSM LTH-) and incubated for 3–5 days at 30°C. For Yeast-2-Hybrid experiments involving phytochromes, phycocyanobilin (Frontier Scientific, Logan, Utah, Cat-No: FSIP14137) was supplemented to the media as indicated in the figure legends and incubation temperature was reduced to 26°C. Light treatment was performed using either 660 nm (1 µmol $m^{-2}$ $s^{-1}$, R) or 740 nm (10 µmol $m^{-2}$ $s^{-1}$, FR) LED panels. Filter lift assays and quantification of β-Gal activity using ONPG were performed as described (*Sheerin et al., 2015*; *Clontech, 2009*). Total protein extraction for immunoblotting was performed as described (*Kushnirov, 2000*).

## Live imaging

For microscopic live imaging of *A. thaliana* seedlings and *N. benthamiana* mesophyll cells, plant material was transferred under green light conditions to microscope slides and mounted in sterile $H_2O$. Microscopic images were acquired using a Zeiss Axioplan 2io mot (Carl Zeiss, Göttingen, Germany) equipped with a Photometrics CoolSnap-HQ 12-bit monochrome CCD camera (Roper Scientific, Tucson, AZ), external filter wheels (LUDL, Hawthorne, NY), and filter sets for mCherry (F36-508, excitation 562 nm, emission 641 nm; AHF Analysentechnik, Tübingen, Germany), YFP (F31-028, excitation 500 nm, emission 515 nm; AHF Analysentechnik, Tübingen, Germany), and CFP (F31-044, excitation 436 nm, emission 455 nm; AHF Analysentechnik, Tübingen, Germany). For FR light treatment, plants were kept for 4–6 hr in FR light (740 nm, 40 µmol $m^{-2}$ $s^{-1}$).

## Total protein extraction

Seedlings were grown for four days under the light conditions indicated in the figure legends. Total protein was extracted by grinding 100 mg of seedlings in liquid $N_2$ followed by the addition of 250 µl of extraction buffer (65 mM Tris/HCl pH 7.3, 4 M Urea, 3% SDS, 10% Glycerol, 0.05% Bromphenol blue, 20 mM DTT, 1× Protease Inhibitor Cocktail (Sigma-Aldrich, Cat-No: I3911)) preheated to 95°C. Soluble protein was separated by centrifugation (15 min, 20,000× *g*) and protein content measured using the Amido black method (*Popov et al., 1975*).

## Immunoblotting

Equal amounts of proteins were separated by 10% SDS-PAGE and transferred to PVDF membrane. Membranes were blocked with 5% skim milk powder in PBS-T (137 mM NaCl, 2.7 mM KCl, 10 mM $Na_2HPO_4$, 1.8 mM $KH_2PO_4$, pH 7.4, 0.5% Tween-20). Membranes were probed with antibodies (see *Supplementary file 3* for details and dilutions). Immunodetection was performed using CDP-Star (Sigma-Aldrich, Cat-No: 11759051001) according to manufacturers instructions. αACT antibodies or amidoblack staining served as loading control.

## AP-MS analysis

Two grams of plant material was collected under respective light conditions and ground for 8 min in liquid $N_2$. Protein was extracted using 4 ml IP buffer (pH 7.8; 134 mM $Na_2HPO_4$, 1.56 mM $NaH_2PO_4$, 150 mM NaCl, 1 mM KCl, 1 mM EDTA, 0.5% Triton X-100, 1 mM $Na_3VO_4$, 2 mM $Na_4P_2O_7$, 10 mM NaF, 1× Protease Inhibitor Cocktail (Sigma-Aldrich, Cat-No: I3911), 1× Protease cOmplete Inhibitor Cocktail (Sigma-Aldrich, Cat-No: 04693159001)). The lysate was further homogenised using Potter-Elvehjem homogenisers and cleared by centrifugation (20,000× *g*, 4°C). A total of 100 µl Anti-GFP MicroBeads (Miltenyi, Cat-No: 130-091-125) were added to the supernatant and incubated for 2 hr in darkness at 4°C. The pulldown was performed as indicated in the manufacturers protocol. Elution was done using 3× 50 µl 0.1 M TEA pH 13.1 and directly neutralised with 3 µl 1 M MES pH 2. Eluate fractions 1, 2, and 3 were combined (approx. 140–150 µl with pH 6–7) and 200 mM ammonium bicarbonate buffer pH 8 was added to obtain a total volume of 200 µl. The disulfide bonds in the proteins were reduced with TCEP and the free sulfhydrils derivatised with MMTS. Proteins were then precipitated with cold acetone (6× volume) overnight at −20°C. The washed pellets were rehydrated in 25 mM ammonium bicarbonate buffer pH 8 at 37°C for 1 hr, then trypsin was added and the proteins digested overnight. An aliquot of the digests was analysed by LC-MS/MS on a Thermo Orbitrap Fusion Lumos mass spectrometer on-line coupled to a Waters nanoAcquity UPLC in data-dependent fashion using HCD fragmentation as follows. Five µl of the sample was loaded onto a Waters Symmetry trap column (C18, 5 µm, 180 µm × 20 mm) in 99% solvent A (0.1% formic acid in water) at a flow rate of 5 µl $min^{-1}$

for 5 min. Peptides were separated by increasing solvent B (0.1% formic acid in acetonitrile) from 5% to 35% in 90 min, then to 50% in 5 min and up to 90% in 4 min. MS survey scans (m/z 380–1580; AGC 400,000; max inject time 50 ms, resolution 120,000) were followed by HCD scans (m/z auto; AGC 50,000; max inject time 100 ms, resolution 15,000; isolation width 1.6 Da; cycle time 2 s, normalised collision energy 35%) on the most abundant multiply charged ions with a minimum intensity threshold of 25,000, then excluded for 10 s. Raw data were processed by Protein Discoverer (v1.4). The resulting peaklists were submitted to database search by ProteinProspector (v5.22.0) first against the full Swissprot 2019.6.12 database (560,292 entries), then against *Arabidopsis thaliana* entries in the Uniprot 2019.6.12 database also considering the YFP-HA and HA-YFP-NOT9B sequences and contaminants identified from the Swissprot database in the initial database search (89,235 entries). A precursor mass tolerance of 5 ppm and a fragment mass tolerance of 20 ppm was used. Only fully tryptic peptides were considered with a maximum of two missed cleavages. Methylthio modification of Cys residues was used as fixed modification and Met oxidation, acetylation of protein N-termini and pyroglutamic acid formation of peptide N-terminal Gln residues as variable modifications. Acceptance criteria were as follows: protein score >22, peptide score >15, protein E < 0.01, peptide E < 0.05, protein FDR < 1%, peptide FDR < 1%, and a minimum of two unique peptide identifications per protein. Peptide search results are summarised in *Supplementary file 1*; raw data are available at MassIVE (ftp://massive.ucsd.edu/MSV000086324/).

## Analysis of cotyledon opening angle

Plants were grown for 4 days under the respective light conditions. At least 17 individual seedlings of each line were measured using the ImageJ software as described elsewhere (*Kretsch, 2010*).

## Light sources

For measurement of hypocotyl elongation, modified Prado 500 W universal projectors (Leitz, Wetzlar, Germany) were used as light sources with Xenophot longlife lamps (Osram, Berlin). Light was passed through narrow-band filters (716 DAL for FR, KG65 for R). For all other experiments 740 nm LEDs have been used for FR light and 656 nm LEDs for R light. For plant cultivation fluorescent bulbs were used. For induction of germination plants were kept in a growth cabinet in W light (Sanyo, Osaka, Japan). For cultivation plants were kept under fluorescent white light. Spectra of all light sources can be found in *Figure 10*.

## Anthocyanin extraction and quantification

Seedlings were grown for four days in either D, FR (740 nm, 40 µmol m$^{-2}$ s$^{-1}$), or B (436 nm, 30 µmol m$^{-2}$ s$^{-1}$ PAR) on ½ MS/1.2% agar supplemented with 1.5% sucrose. Anthocyanin was extracted by collecting 25 seedlings from each treatment and genotype into 500 µl extraction buffer (18% (v/v) 1-propanol, 0.37% (v/v) HCl). Samples were heated for 2 min to 95℃, chilled on ice for 5 min, and incubated overnight at 4℃ under constant shaking in darkness. Plant material was separated by centrifugation for 10 min (13,000× *g*, room temperature) and the supernatant was analysed. Anthocyanin was quantified by measuring A$_{535}$ and A$_{650}$ using a plate reader. The relative amount (A$_{535}$-A$_{650}$) was calculated.

## Transient transformation of *Nicotiana benthamiana*

Transient transformation of tobacco (*N. benthamiana*) was performed as described (*Chapman et al., 2004*). For all experiments *p35S:P19* was coinfiltrated.

## Stable transformation of *Arabidopsis thaliana*

Transformation of *A. thaliana* using the floral dip method was performed as described (*Clough and Bent, 1998*).

## Live imaging of luciferase activity

For in vivo detection of luciferase activity, seedlings were grown for 4 days under indicated light conditions. For detection seedlings were sprayed with luciferin solution (2 mM D-Luciferin (Biosynth #L-8220), 0.001% Triton-X, 100 mM Tris-Phosphate pH 8). Pictures of the luminescence were taken

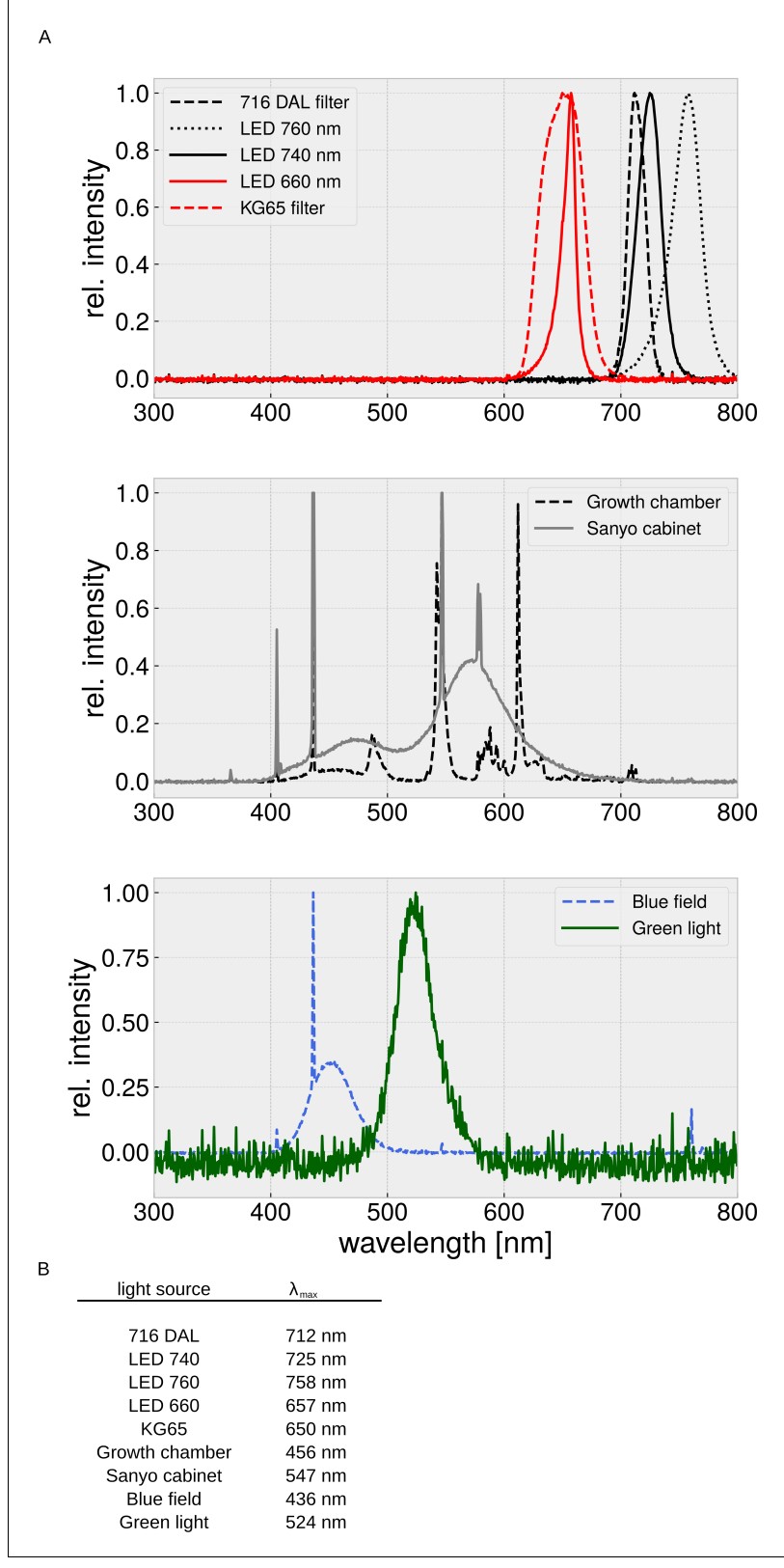

**Figure 10.** Emission spectra of light sources. (**A**) Spectra of light sources used in this study were characterised using a photospectrometer. Emission in the wavelength range from 300 to 800 nm was normalised to emission at $\lambda_{max}$. (**B**) $\lambda_{max}$ for light sources used in this study.

using a cooled CCD-camera. False colour images were generated using the ImageJ software package.

For in vivo detection of luciferase activity in yeast cells, plates were sprayed with luciferin solution (2 mM D-Luciferin (Biosynth #L-8220), 0.001% Triton-X, 100 mM Tris-Phosphate pH 8). Pictures of the luminescence were taken using a cooled CCD-camera. False colour images were generated using the ImageJ software package.

## GUS stain of promoter:GUS reporter line

Staining of *pNOT9B:GUS* transgenic seedlings was done as described (*Enderle et al., 2017*).

## Image processing

Blots and microscopic images were brightness/contrast adjusted and cropped using ImageJ v1.52a and assembled using Inkscape 0.92.4. In some figures, blots were cropped for spacing reasons; dotted lines show cropping.

## Statistical analysis

Statistical analysis was performed as indicated in the figure legends. Plots were created using the Matplotlib package in Python 3.7.6 using Spyder IDE v4.0.1.

## GO term analysis

Identifiers have been parsed to the PANTHER Classification system and analysed towards the GO aspect 'molecular function'. *p*-value cut-off was set to p<0.01.

## Sequence alignment

Sequences were retrieved from UniProt and aligned in JalView using MAFFT with default settings (*UniProt Consortium, 2019*; *Katoh and Standley, 2013*; *Waterhouse et al., 2009*). UniProt sequence identifiers are included in the sequence names.

## TAIR accessions identifiers for genes mentioned in this study

ACT1, AT2G37620; *AGO1*, AT1G48410; *CAF1A*, AT3G44260; *CAF1B*, AT5G22250; *CCR4A*, AT3G58560; *CHS*, AT5G13930; *COP1*, AT2G32950; *DCL1*, AT1G01040; *DCP1*, AT3G13300; *ELIP1*, AT3G22840; *ELIP2*, AT4G14690; *EPR1*, AT1G18330; *FHL*, AT5G02200; *FHY1*, AT2G37678; *HEN1*, AT4G20910; *HY5*, AT5G11260; *HYL1*, AT1G09700; *MYBD*, AT1G70000; *NOT1*, AT1G02080; *NOT2B*, AT5G59710; *NOT4A*, AT5G60170; *NOT4B*, AT3G45630; *NOT9A*, AT3G20800; *NOT9B*, AT5G12980; *NOT9C*, AT2G32550; *PHYA*, AT1G09570; *PHYB*, AT2G18790; *PIF4*, AT2G43010; *RH8*, AT4G00600; *RRC1*, AT5G25060; *SE*, AT2G27100; *SFPS*, AT1G30480; *SPA1*, AT2G46340; *XRN4*, AT1G54490.

## Acknowledgements

This study was supported by the German Research Foundation (DFG) under Germany's Excellence Strategy (BIOSS – EXC-294, project C6 to AH; the GSC-4/Spemann Graduate School to PS and KLL; CIBSS – EXC-2189 – Project ID 390939984, project C1 to AH), HI 1369/10–1 (Project ID 453030721) to AH, HO 2793–3 to UH, and Germany's Excellence Strategy–EXC 2048/1, Project 390686111, to UH, and in part by the Ministry for Science, Research, and Arts of the State of Baden-Wuerttemberg. The work in Hungary was supported by the Hungarian Scientific Research Fund (OTKA, K-132633) and the Economic Development and Innovation Operative Program (GINOP-2.3.2-15-2016-00001, GINOP-2.3.2-15-2016-00015, and GINOP-2.3.2-15-2016-00032 to EK, AV, and KFM). AMS was supported by a PhD fellowship (Project Reference 14546015) from the Luxembourg National Research Fund (FNR). We are grateful to Martina Krenz and Tessy Albonetti (University of Freiburg, Germany) for technical assistance, Prof. Thomas Laux (University of Freiburg, Germany) for providing *ago1-27* seeds, Dr. Maximilian Ulbrich (University of Freiburg, Germany) for providing pcDNA3.1-g2mGFP, the staff of the Core Facility Signalling Factory (Signalhaus, University of Freiburg) for providing and maintaining the HEK293T cell line, and the Nottingham Arabidopsis Stock Centre (NASC) for providing *not9b-1* and *not9b-2* T-DNA insertion mutant lines. pGGF012 was a gift from Jan Lohmann

(Addgene plasmid # 48849; http://n2t.net/addgene:48849; RRID:Addgene_48849). The article processing charge was funded by the University of Freiburg in the funding programme Open Access Publishing.

## Additional information

### Funding

| Funder | Grant reference number | Author |
|---|---|---|
| Deutsche Forschungsgemeinschaft | EXC-294 | Andreas Hiltbrunner |
| Deutsche Forschungsgemeinschaft | EXC-2189 | Andreas Hiltbrunner |
| Deutsche Forschungsgemeinschaft | GSC-4 | Philipp Schwenk<br>Klara L Lesch |
| Deutsche Forschungsgemeinschaft | HO 2793-3 | Ute Hoecker |
| Deutsche Forschungsgemeinschaft | HI 1369/10-1 | Andreas Hiltbrunner |
| Hungarian Scientific Research Fund | K-132633 | Katalin F Medzihradszky<br>Eva Klement<br>András Viczián |
| Economic Development and Innovation Operative Program | GINOP-2.3.2-15-2016-00001 | Katalin F Medzihradszky<br>Eva Klement<br>András Viczián |
| Economic Development and Innovation Operative Program | GINOP-2.3.2-15-2016-00015 | Katalin F Medzihradszky<br>Eva Klement<br>András Viczián |
| Economic Development and Innovation Operative Program | GINOP-2.3.2-15-2016-00032 | Katalin F Medzihradszky<br>Eva Klement<br>András Viczián |
| Fonds National de la Recherche Luxembourg | 14546015 | Anne-Marie Staudt |

The funders had no role in study design, data collection and interpretation, or the decision to submit the work for publication.

### Author contributions

Philipp Schwenk, Conceptualization, Resources, Data curation, Funding acquisition, Investigation, Visualization, Methodology, Writing - original draft, Writing - review and editing; David J Sheerin, Resources, Investigation, Writing - review and editing; Jathish Ponnu, Conceptualization, Investigation, Visualization, Writing - original draft, Writing - review and editing; Anne-Marie Staudt, Klara L Lesch, Elisabeth Lichtenberg, Investigation, Writing - review and editing; Katalin F Medzihradszky, Ute Hoecker, Conceptualization, Funding acquisition, Writing - review and editing; Eva Klement, Conceptualization, Data curation, Investigation, Writing - original draft, Writing - review and editing; András Viczián, Conceptualization, Funding acquisition, Investigation, Writing - review and editing; Andreas Hiltbrunner, Conceptualization, Supervision, Funding acquisition, Investigation, Writing - original draft, Project administration, Writing - review and editing

### Author ORCIDs

Jathish Ponnu (iD) http://orcid.org/0000-0002-3276-7068
András Viczián (iD) https://orcid.org/0000-0003-2055-3430
Andreas Hiltbrunner (iD) https://orcid.org/0000-0003-0438-5297

### Decision letter and Author response

Decision letter https://doi.org/10.7554/eLife.63697.sa1

Author response https://doi.org/10.7554/eLife.63697.sa2

## Additional files

### Supplementary files

• Supplementary file 1. This file contains the complete list of proteins identified in the AP-MS experiment described in *Figure 7—figure supplement 1*. Protein hits with at least two unique peptide matches in any of the samples were included in this list.

• Supplementary file 2. This file contains detailed information on plasmids used in this study, including the cloning procedure and sequences of primers and gBlock fragments used for cloning. This file also contains detailed information on mutants and transgenic lines used in this study, including the generation of transgenic lines and genotyping of mutants.

• Supplementary file 3. This file contains detailed information on primers and probes used for qPCR analyses as well as on antibodies used for immunblotting and/or CoIPs.

• Transparent reporting form

### Data availability

The authors declare that all data supporting the findings of this study are available within the paper and its supplementary information files. Raw data from AP-MS are available at MassIVE (ftp://massive.ucsd.edu/MSV000086324/).

The following dataset was generated:

| Author(s) | Year | Dataset title | Dataset URL | Database and Identifier |
| --- | --- | --- | --- | --- |
| Klement E | 2020 | NOT9 Co-IP in Arabidopsis thaliana | https://doi.org/10.25345/C5SR3C | MassIVE, 10.25345/C5SR3C |

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
