## [Decision Letter]

**Acceptance summary:**

The revised version of your manuscript provides additional evidences that support the existence of a direct regulatory link between phyA and CCR4-NOT complex component NOT9B to regulate far-red (FR) light signaling in Arabidopsis. According to the proposed model, NOT9B acts as a negative regulator of early seedling development, acting downstream of phyA. Subcellular localization and protein-protein interaction assays show that such a function occurs mainly at the nucleus, where active phyA competes with NOT1, a central component of CCR4-NOT complexes, for interaction with NOT9B. As a result, NOT9B is recruited into photobodies together with phyA, which likely impairs NOT9B function in the control of alternative splicing and FR light-regulated gene expression.

In sum, this study describes a novel function for phyA as a repressor of CCR4-NOT complex component NOT9B, a negative regulator of FR light signaling in Arabidopsis.

**Decision letter after peer review:**

Thank you for submitting your article "Unexpected function of the CCR4-NOT complex in phytochrome A-mediated light signalling in plants" for consideration by *eLife*. Your article has been reviewed by three peer reviewers, including Vicente Rubio as the Reviewing Editor and Reviewer #1, and the evaluation has been overseen Christian Hardtke as the Senior Editor.

Essential Revisions:

Whereas the data provided convincingly show a functional connection between phyA and NOT9B to regulate seedling development in response to FR light, much less is known on how NOT9B controls gene expression and developmental responses downstream of phyA. It is proposed, based on phenotypic similarities between *not9b* and *ago1* mutants and affinity copurification of both NOT9B and AGO1, that AGO1 could mediate, at least partly, NOT9B functions in FR light signaling. Genetic analysis of the effect of *ago1* mutation in NOT9B overexpressing lines would be highly desirable to evaluate the contribution of AGO1 to NOT9B-mediated repression of FR responses. However, such a line might not be available at this point, precluding its characterization in a timely manner. A feasible alternative is to analyze the transcriptomic behavior of lines with altered NOT9B and AGO1 function (already available), by means of simple RNA seq analyses, both under dark and FR light conditions. Such RNAseq analysis would be also very helpful to assess the extent to which the gene expression changes reflect the complex phenotypic changes that the authors show. In some cases, the mutant and OX lines display similar phenotypes. The authors provide explanations for this, but a comprehensive gene expression analysis as opposed to choosing a limited number of genes would give a broader picture of the visible and molecular phenotypes regulated by these complexes. Moreover, these analyses might help to identify additional alternative splicing events controlled by the phyA-NOT9B-AGO1 module during FR light signaling, which is limited to a single gene in this study.

Additionally, a key observation in the proposed model is the mutually-exclusive binding of NOT9B to NOT1 and phyA that is regulating early events in gene expression. Using yeast three-hybrid and co-overexpression, convincing evidence is provided for this mechanism. Additional support would be showing that co-overexpression of NOT9B and NOT1M, if such a line is available, also ameliorates the gene expression defects of the NOT9Box line as shown in Figure 9A, since the gene expression effects of excess NOT9B should be squelched by excess NOT1M as it is shown for hypocotyl length (Figure 6).

Finally, for a protein to function analysis in such a large complex, it is difficult to assess pairwise interaction using yeast-two-hybrid or in vivo immunoprecipitation assays as these proteins might be in a complex and show interaction even in yeast. The authors can modify their interaction claims or add additional in vitro interactions assays using purified proteins.

With regard to the title, this study unveils an unknown mechanism by which phyA regulates the function of downstream factors acting at the nucleus to control RNA metabolism and gene expression in response to FR light. The use of the term "unexpected" (as in unanticipated) might not be the appropriate term to describe it, despite the novelty and interest of the findings. The authors might will to provide an alternative title (i.e. "Uncovering a novel function…").

---

## [Author Response]

Essential Revisions:Whereas the data provided convincingly show a functional connection between phyA and NOT9B to regulate seedling development in response to FR light, much less is known on how NOT9B controls gene expression and developmental responses downstream of phyA. It is proposed, based on phenotypic similarities between not9b and ago1 mutants and affinity copurification of both NOT9B and AGO1, that AGO1 could mediate, at least partly, NOT9B functions in FR light signaling. Genetic analysis of the effect of ago1 mutation in NOT9B overexpressing lines would be highly desirable to evaluate the contribution of AGO1 to NOT9B-mediated repression of FR responses. However, such a line might not be available at this point, precluding its characterization in a timely manner. A feasible alternative is to analyze the transcriptomic behavior of lines with altered NOT9B and AGO1 function (already available), by means of simple RNA seq analyses, both under dark and FR light conditions. Such RNAseq analysis would be also very helpful to assess the extent to which the gene expression changes reflect the complex phenotypic changes that the authors show. In some cases, the mutant and OX lines display similar phenotypes. The authors provide explanations for this, but a comprehensive gene expression analysis as opposed to choosing a limited number of genes would give a broader picture of the visible and molecular phenotypes regulated by these complexes. Moreover, these analyses might help to identify additional alternative splicing events controlled by the phyA-NOT9B-AGO1 module during FR light signaling, which is limited to a single gene in this study.

We agree that this is an interesting experiment and we plan to do it in the future, yet it is not possible at the moment and would considerably delay publication of our current manuscript. The take home message of the current manuscript is that NOT9B and the CCR4-NOT complex play a role in light signalling and that phyA can regulate CCR4-NOT, which is novel and interesting as acknowledged by the reviewers. As the conclusions of the manuscript do not depend on this experiment, you have previously agreed that this experiment is not required for the revised version of the manuscript to be accepted for publication for which we are very grateful.

Additionally, a key observation in the proposed model is the mutually-exclusive binding of NOT9B to NOT1 and phyA that is regulating early events in gene expression. Using yeast three-hybrid and co-overexpression, convincing evidence is provided for this mechanism. Additional support would be showing that co-overexpression of NOT9B and NOT1M, if such a line is available, also ameliorates the gene expression defects of the NOT9Box line as shown in Figure 9A, since the gene expression effects of excess NOT9B should be squelched by excess NOT1M as it is shown for hypocotyl length (Figure 6).

The reviewers appreciate that data shown in the manuscript provide convincing evidence for competition between NOT9B and phyA for binding NOT1. We also showed that the long hypocotyl phenotype of NOT9Box lines is partially rescued by overexpression of NOT1 M. The reviewers suggested to further confirm this (partial) complementation by testing for complementation of gene expression defects of NOT9Box by overexpression of NOT1 M. We performed this experiment and data presented in Figure 6—figure supplement 5 show that overexpression of NOT1 M partially rescues the NOT9Box gene expression defect for representative light marker genes (*ELIP1*, *ELIP2*, *CHS*). Expression levels for all three genes are still lower in NOT9Box NOT1 Mox double transgenic seedlings than in the wildtype control, but expression levels are significantly higher than in the NOT9Box line not overexpressing the NOT1 M transgene. This finding is fully consistent with the partial rescue of the hypocotyl growth phenotype of NOT9Box by overexpression NOT1 M.

Finally, for a protein to function analysis in such a large complex, it is difficult to assess pairwise interaction using yeast-two-hybrid or in vivo immunoprecipitation assays as these proteins might be in a complex and show interaction even in yeast. The authors can modify their interaction claims or add additional in vitro interactions assays using purified proteins.

To provide further evidence for phyA/NOT9B complex formation, we performed CoIP assays from HEK293T cells co-expressing phyA and NOT9B. The HEK293T CoIP data presented in Figure 1—figure supplement 4 confirm that phyA and NOT9B are in complex. Y2H and HEK293T CoIP data show that no additional plant-specific proteins are required for phyA/NOT9B complex formation but we cannot formally rule out the possibility that components of the CCR4-NOT complex conserved in animals, yeast, and plants (and therefore present in yeast and HEK293T cells) bridge between NOT9B and phyA in these assays. Yet, if phyA would bind indirectly to NOT9B through other components of the CCR4-NOT complex, we would expect complex formation for phyA and both NOT9A and NOT1 in the Y2H assay and recruitment of NOT9A and NOT1 into phyA-dependent photobodies. However, we did not observe either. Nevertheless, we agree with the reviewers that we cannot formally exclude that components of the CCR4-NOT complex might bridge between phyA and NOT9B in the Y2H or CoIP assays, which we point out in the Discussion of the revised version of the manuscript. Furthermore, we also mention that testing complex formation of heterologously expressed phyA and NOT9B could prove a direct interaction. However, both phyA and NOT9B are very difficult to express in *E. coli* and therefore such an experiment is beyond the scope of this manuscript. Instead, we have modified the interaction statements in the manuscript as suggested by the reviewers.

With regard to the title, this study unveils an unknown mechanism by which phyA regulates the function of downstream factors acting at the nucleus to control RNA metabolism and gene expression in response to FR light. The use of the term "unexpected" (as in unanticipated) might not be the appropriate term to describe it, despite the novelty and interest of the findings. The authors might will to provide an alternative title (i.e. "Uncovering a novel function…").

We changed the title of the manuscript according to the reviewers suggestion. The title of the revised manuscript is “Uncovering a novel function of the CCR4-NOT complex in phytochrome A-mediated light signalling in plants”.